# Description of two new species and a new population of *Mesobiotus* cf. *coronatus* from Cotacachi-Cayapas National Park, Ecuador

Anastasiia Polishchuk[1]*, Pushpalata Kayastha[2], Dominika Młodzianowska[3], Martyna Michalska[3], Magdalena Gawlak[4], Jędrzej Warguła[1], Łukasz Kaczmarek[1]

1 Department of Animal Taxonomy and Ecology, Adam Mickiewicz University in Poznań, Poznań, Poland, 2 Biology Department, The University of North Carolina at Chapel Hill, Chapel Hill, North Carolina, United States of America, 3 Department of Genetics and Biosystematics, Faculty of Biology, University of Gdańsk, Gdańsk, Poland, 4 Institute of Plant Protection – National Research Institute, Poznań, Poland

* anastasiia.m.polishchuk@gmail.com

## Abstract

In this study, we present descriptions of two new eutardigrade species and additionally a detailed description of new population of *Mesobiotus* cf. *coronatus* discovered in the Cotacachi-Cayapas National Park in Ecuador. The presented analysis is based on morphological and morphometric data using light and scanning electron microscopy; description of *Macrobiotus sharopovi* sp. nov. and the population of *Mesobiotus* cf. *coronatus* were also supported with genetic data (nuclear barcode sequences, i.e., 18S rRNA, 28S rRNA and ITS-2, and one mitochondrial COI sequence). Based on the egg morphology, *Mac. sharopovi* sp. nov. belongs to *Mac. paulinae* morpho-group and is most similar to *Mac. papei, Mac. paulinae, Mac. polypiformis* and *Mac. shonaicus* but differs from them mainly by some characters of eggs (number of egg processes on circumference, details of eggshell morphology and features of filaments and their number) and adults (size of cuticular pores). Based on such features as the presence of gibbosities and well visible dorsal sculpture, the second new species, i.e., *Ramazzottius syraxi* sp. nov. is the most similar to *Ram. baumanni* species complex, *Ram. belubellus, Ram. saltensis* and *Ram. szeptycki* but differs from them by gibbosities configuration and morphology of dorsal sculpture. *Mesobiotus* cf. *coronatus*, which belongs to *Meb. harmsworthi* morpho-group, is potentially a new species but due to unclear taxonomic position of nominal *Meb. coronatus*, a formal description of this taxon is not possible in the present study.

## Introduction

The Cotacachi-Cayapas National Park, established in 2019, originated from the Cotacachi-Cayapas Ecological Reserve, which was erected in 1968. It is located in the

**Data availability statement:** All relevant data are within the manuscript and its Supporting Information files.

**Funding:** The work of Anastasiia Polishchuk was supported by grant no. 2022/01/4/NZ4/00009 from the National Science Centre (Poland). The funders had no role in study design, data collection and analysis, decision to publish, or preparation of the manuscript.

**Competing interests:** The authors have declared that no competing interests exist.

northwestern part of Ecuador and spans the provinces of Esmeraldas (San Lorenzo, Eloy Alfaro and Río Verde Cantons) and Imbabura (Cotacachi, Urcuquí and Ibarra Cantons), with a total area of 260,961 ha [1]. It is one of the largest protected areas in the country with exceptional biodiversity and varied ecosystems. It extends from the high-altitude Andean region covered by tundra-like vegetation (páramo) with the notable geographical feature Cotacachi Volcano (4989 m asl) on the west to the lowland tropical rainforests with swamps and lagoons on the east. That eastern part belongs to the Chocó biogeographic region, which in turn is a part of the Tumbes-Chocó-Magdalena biodiversity hotspot, one of the world's most species-rich areas [2].

The variety of natural environments and the high-altitude range in the park contribute to the existence of a high diversity of species of flora and fauna, e.g., 2225 species of plants, 139 species of mammals, 845 bird species, 111 reptiles, 124 amphibians and 39 fish species with showcasing a high level of endemism [1]. At the same time, Cotacachi-Cayapas National Park is a focus for conservationists due to the threats of deforestation, mining and climate change, especially the rainforests of the Chocó region [2,3].

The phylum Tardigrada includes over 1400 described species of microscopic invertebrates [4] noted for their exceptional resilience across diverse environmental conditions [5,6]. These organisms inhabit terrestrial, freshwater and marine ecosystems worldwide, and can be found in mosses, lichens, soil, leaf litter, as well as in/on marine and freshwater sediments and plants [7].

The family Macrobiotidae Thulin 1928 [8], consisting of limno-terrestrial tardigrades, encompasses 14 recognized genera [9]. Notably, a significant portion of species diversity within this family is concentrated in four genera: *Macrobiotus* Schultze 1834 [10], *Mesobiotus* Vecchi, Cesari, Bertolani, Jönsson, Rebecchi & Guidetti 2016 [11], *Minibiotus* Schuster 1980 (in Schuster et al. 1980 [12]) and *Paramacrobiotus* Guidetti, Schill, Bertolani, Dandekar & Wolf 2009 [13]. This classification reflects the evolving understanding of their phylogenetic relationships and ecological significance.

The genus *Macrobiotus*, established by Schultze [10], has one of the most complex revision histories within the phylum Tardigrada, reflecting its extensive phenotypic diversity, especially in egg ornamentation morphology [9,14–16]. This complexity has prompted numerous studies aimed at refining the definition of the genus and clarifying the classification of smaller groups of species within it [9,15–21]. Current studies support the internal division of *Macrobiotus* into three distinct phylogenetic lineages – clades A, B, and C with monophyletic groups, species complexes (*Mac. pallarii*, *Mac. pseudohufelandi*, *Mac. ariekammensis* and *Mac. polonicus-persimilis* complexes) and non-monophyletic species morpho-groups (including *Mac. hufelandi* and *Mac. nelsonae* morpho-groups) [9,16,19–21]. Despite these categorizations, the subdivision of *Macrobiotus* remains complex, prompting Kaczmarek et al. [16] to propose an artificial division of all taxa into 12 morphological groups based solely on eggshell morphology, potentially simplifying taxonomic research and facilitating better navigation among taxa, although the long-term acceptance and utility of this methodology remains to be seen.

The genus *Mesobiotus* encompasses 82 nominal species, including four designated as *nomina inquirenda* [4,22,23]. Established by Vecchi et al. [11], *Mesobiotus* is supported by both morphological and genetic evidence. Research indicates that the genus is monophyletic; however, it does not support the division into the two traditionally recognized species groups – *harmsworthi* and *furciger* – which are distinguished primarily by the morphology of egg processes (specifically, the *furciger* group is characterized by dichotomous branching at the tips of the egg processes), and due to the lack of reciprocal monophyly, Short et al. [24] suggested that these informal groups should be discontinued. In turn, Stec [25] advocates for retaining these morpho-groups for their utility in taxonomic identification and communication, and introduced a new morpho-group – *montanus*, characterized by dome-shaped egg processes, to enhance classification accuracy and clarity in taxonomic discourse.

In the family Ramazzottiidae Sands, McInnes, Marley, Goodall-Copestake, Convey & Linse 2008 [26], genus *Ramazzottius* Binda & Pilato 1986 [27] encompasses 30 species [4], including the type species *Ramazzottius oberhaeuseri* (Doyère, 1840) [28], and characterized by the presence of extremely different in size and shape internal and external claws, apophyses for the insertion of the stylet muscles in the shape of "blunt hooks" and general presence of a paired elliptical organs on the head [27,29]. Species of this genus are often found in xeric mosses and lichens and are one of the most common and distributed globally eutardigrades [30–34]. Despite this, the genus *Ramazzottius* includes various species complexes with many taxonomic problems and not fully resolved phylogenetic relationships (e.g., *Ram. oberhaeuseri*, *Ram. baumanni* and *Ram. szeptyckii* morpho-groups) [35–38].

The tardigrade fauna of Ecuador is one of the poorest studied not only in South America, but also in the world. So far, only 27 species were reported from mainland Ecuador [32,39–42] and many of them are now considered as species complexes or doubtful taxa (for more details see, e.g., Kaczmarek et al. [32]). This suggests that tardigrades in this region are very poorly known, and many taxa await discovery.

In this paper, we are describing two new eutardigrade species: *Macrobiotus sharopovi* sp. nov. (*Mac. hufelandi* morpho-group) and *Ramazzottius syraxi* sp. nov., and the new population of *Mesobiotus* cf. *coronatus*.

## Materials and methods

### Samples and sample processing

Forty-four samples of mosses, lichens, cryptograms and algae were collected from soil, trees, rocks, shrubs, dead wood and concrete wall on December 16–17, 2014 by Milena Roszkowska, Pedro Rios Guayasamín and Łukasz Kaczmarek in the south of the Cotacachi-Cayapas National Park (Imbabura Province, Ecuador) (permit №001–15IC-FLO-FAU-DNB/MA, issued by the Ministry of Agriculture, Livestock Aquaculture and Fisheries of Ecuador). Additional information regarding the ethical, cultural, and scientific considerations specific to inclusivity in global research is included in the Supporting Information (S1 Checklist).

Samples were packed in paper envelopes, dried at a temperature of *ca.* 20°C and delivered to the laboratory at the Faculty of Biology, Adam Mickiewicz University in Poznań, Poland. Tardigrades and their eggs were extracted from the samples and studied following the protocol in Stec et al. [43].

### Microscopy and imaging

In total 157 specimens and 44 eggs (97 specimens and 14 eggs of *Mac. sharopovi* sp. nov., 47 specimens and 30 eggs of *Meb.* cf. *coronatus* and 13 specimens of *Ram. syraxi* sp. nov. (seven specimens from the type locality and six specimens from the additional locality)) were mounted on microscope slides in Hoyer's medium and secured with a cover slips. Then slides were dried in the heater for two days at *ca.* 60°C and sealed with transparent nail polish. Prepared slides were examined with an Olympus BX41 phase-contrast light microscope (PCM) associated with an ART-CAM–300Mi digital camera (Olympus Corporation, Shin-juku–ku, Japan) for imaging.

An additional 38 specimens and 10 eggs of *Mac. sharopovi* sp. nov. and 25 specimens and 10 eggs of *Meb.* cf. *coronatus* were prepared for Scanning Electron Microscope (SEM) analysis according to the protocol in Roszkowska et al. [41] and examined under high vacuum in Hitachi S3000N SEM (Hitachi, Japan).

All figures were assembled in Corel Photo-Paint 2021. For deep structures that could not be fully focused on a single photograph, a series of 2–10 images were taken every *ca.* 0.5 micrometers [µm] and then assembled into a single deep-focus image manually in the above-mentioned program.

## Morphometrics and morphological nomenclature

All measurements are given in micrometers [µm]. Structures were measured only if their orientation was suitable. Body length was measured from the anterior extremity to the end of the body, excluding the hind legs.

For *Macrobiotus* and *Ramazzottius*, the classification of the buccal apparatus and claws follows the system developed by Pilato and Binda [29] and amended for *Mesobiotus* by Vecchi et al. [11]. Terminology for the oral cavity armature (OCA) and eggshell morphology is in accordance with Michalczyk and Kaczmarek [44], Stec et al. [37] and Kaczmarek and Michalczyk [15]. Additionally, the sequence of macroplacoid lengths is determined based on Kaczmarek et al. [45], while the morphological states of the cuticular bars on legs are categorized following Kiosya et al. [46]. All other measurements and the nomenclature employed adhere to the guidelines provided by Beasley et al. [47], Stec et al. [37] and Kaczmarek and Michalczyk [15].

The *pt* index, which is defined as the ratio of the length of a given structure to the length of the buccal tube expressed as a percentage according to Pilato [48].

The measurements were performed and recorded according to the "Parachela" using the "Parachela" ver. 1.8 template available from the Tardigrada Register [49] with modification by Massa et al. [17] for Thorpe's Normalization. Raw morphometric data for the specimens of all new described species are given in Supporting Information (S1–S4 Tables).

Genera abbreviations follow Perry et al. [50]

## DNA extraction and genotyping

Twenty specimens of *Mac. sharopovi* sp. nov. and seven specimens of *Meb.* cf. *coronatus* were prepared for genetic analyses, out of which barcodes from three specimens of *Mac. sharopovi* sp. nov. and one specimen of *Meb.* cf. *coronatus* were obtained in good quality and used for analysis.

Light microscopy (LM) was used for *in vivo* specimen identification prior to genomic DNA extraction. In order to obtain voucher specimens, DNA extractions from individuals were performed using a Chelex® 100 resin (Bio-Rad) extraction method supplied by Casquet et al. [51], with modifications outlined in Stec et al. [43]. Exoskeletons were mounted on microscope slides with Hoyer's medium. Four DNA fragments were attempted to be sequenced: one mitochondrial (COI) and three nuclear (18S rRNA, 28S rRNA, and ITS2) using following primers: HCO2198 (5′–TAAACTTCAGGGTGACCAAAAAATCA–3′) and LCO1490 (5′–GGTCAACAAATCATAAAGATATTGG–3′) [52] for the *cox1* gene fragment; SSU01_F (5′–AACCTGGTTGATCCTGCCAGT–3′) and SSU82_R (5′–TGATCCTT CTG-CAGGTTCACCTAC–3′) [26] for the 18S rRNA gene fragment; 28SF0001 (5′–ACCCvCynAATTTAAGCATAT–3′) and 28SR 0990 (5′–CCTTGGTCCGTGTTTCAAGAC–3′) [53] for the 28S rRNA gene fragment; ITS-2 Eutar_Ff (5′– CGTAACGTGAATTGCAGGAC–3′) and ITS-2 Eutar_Rr (5′– TCCTCCGCTTATTGATATGC–3′) [54] for the ITS-2 gene fragment. All the molecular markers underwent amplification using protocol by Kaczmarek et al. [55]. The PCR products were purified with thermosensitive Exonuclease I and FastAP Alkaline Phosphatase (Thermo Scientific) and sequenced with BigDye Terminator v3.1 on an ABI Prism 3130XL Analyzer (Applied Biosystems) according to manufacturer's instructions. Sequence chromatograms were checked for accuracy using FinchTV 1.3.1 (Geospiza Inc.)

 

For comparative molecular analysis, sequence homology and identity were confirmed using the Basic Local Alignment Search Tool (BLAST) [56]. Sequence alignments were performed using the AUTO method for COI and ITS-2 markers, and the Q-INS-I method for ribosomal markers (18S rRNA and 28S rRNA) in MAFFT version 7 [57,58]. The alignments were then manually verified for non-conserved regions in BioEdit. After alignment, sequences were trimmed to specific lengths for each marker, and uncorrected pairwise distances (p-distances) were calculated using MEGA X software [59]. The resulting distance matrices are included in Supporting Information (S5–S6 Tables).

### Nomenclatural Acts

The electronic edition of this article conforms to the requirements of the amended International Code of Zoological Nomenclature, and hence the new names contained herein are available under that Code from the electronic edition of this article. This published work and the nomenclatural acts it contains have been registered in ZooBank, the online registration system for the ICZN. The ZooBank LSIDs (Life Science Identifiers) can be resolved and the associated information viewed through any standard web browser by appending the LSID to the prefix "http://zoobank.org/". The LSID for this publication is: urn:lsid:zoobank.org:pub:E6717E0A-EA1B-49CB-89ED-25BC8E372610. The electronic edition of this work was published in a journal with an ISSN, and has been archived and is available from the following digital repositories: PubMed Central, LOCKSS.

## Results

### Taxonomic account of the new species

Phylum: Tardigrada Doyère 1840 [28]
Class: Eutardigrada Richters 1926 [60]
Order: Parachela Schuster et al. 1980 [12]
Superfamily: Macrobiotoidea Thulin 1928 [8]
Family: Macrobiotidae Thulin 1928 [8]
Genus: *Macrobiotus* Schultze 1834 [10]
*Macrobiotus sharopovi* sp. nov. Polishchuk, Kayastha, Młodzianowska, Warguła & Kaczmarek
urn:lsid:zoobank.org:act:EC6E131E-7E13-4AD6-B801-F40EE1A01A44
(Figs 1–4, Tables 1–2)

Type Locality: Ecuador, Cotacachi-Cayapas National Park, Imbabura Province (00°19'47''N, 78°23'33''W; 3464 m asl), Zona de Intag; cryptograms on shrubs, 17th December 2014, leg. Milena Roszkowska, Pedro Rios Guayasamín and Łukasz Kaczmarek.

Material examined: In total, 179 specimens (155 animals and 24 eggs) were examined. The 97 animals and 14 eggs were mounted on microscope slides in Hoyer's medium, 38 animals and 10 eggs were fixed on SEM stubs and 20 animals were processed for DNA sequencing, out of which we found good quality barcodes from three specimens.

Type depositories: The holotype (1252-25) with 86 paratypes and nine eggs (slides: 1252-*, where the asterisk can be substituted by any of the following numbers: 1, 3, 4, 5, 9, 10, 14, 15, 16, 17, 18, 20, 22, 23, 24, 25, 26, 27 (animals) and 7, 8 (eggs)) are deposited at the Department of Animal Taxonomy and Ecology, Institute of Environmental Biology, Adam Mickiewicz University in Poznań, Uniwersytetu Poznańskiego 6, 61–614 Poznań, Poland. Ten paratypes and five eggs (slides: 1252-*, where the asterisk can be substituted by any of the following numbers: 6 (animals) and 28 (eggs)) are deposited in Institute of Systematics and Evolution of Animals, Polish Academy of Sciences, Sławkowska 17, 31–016, Kraków, Poland.

Etymology: The authors dedicated the species name to Ukrainian scientist Dr. Bizhan Sharopov, neurophysiologist, activist and soldier who tragically died in action defending Ukraine during the Russia–Ukraine War.

 

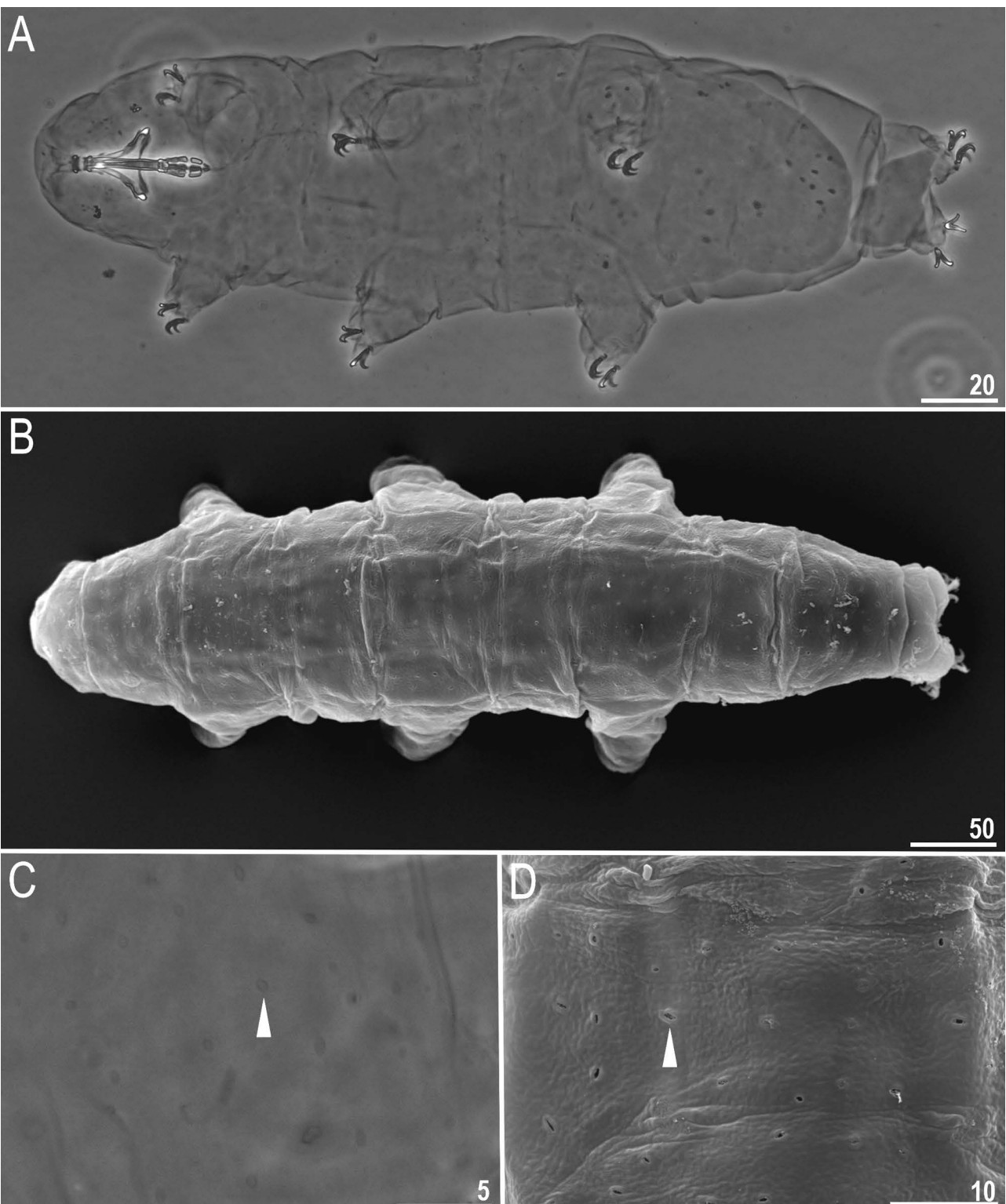

**Fig 1. *Macrobiotus sharopovi* sp. nov.: A – body, ventral view (holotype, PCM); B – body, dorsal view (paratype, SEM); C–D – cuticular pores on dorsal side of the body (paratype, PCM and SEM, respectively).** Filled unindented arrowhead represents cuticular pores. Scale bars in µm.

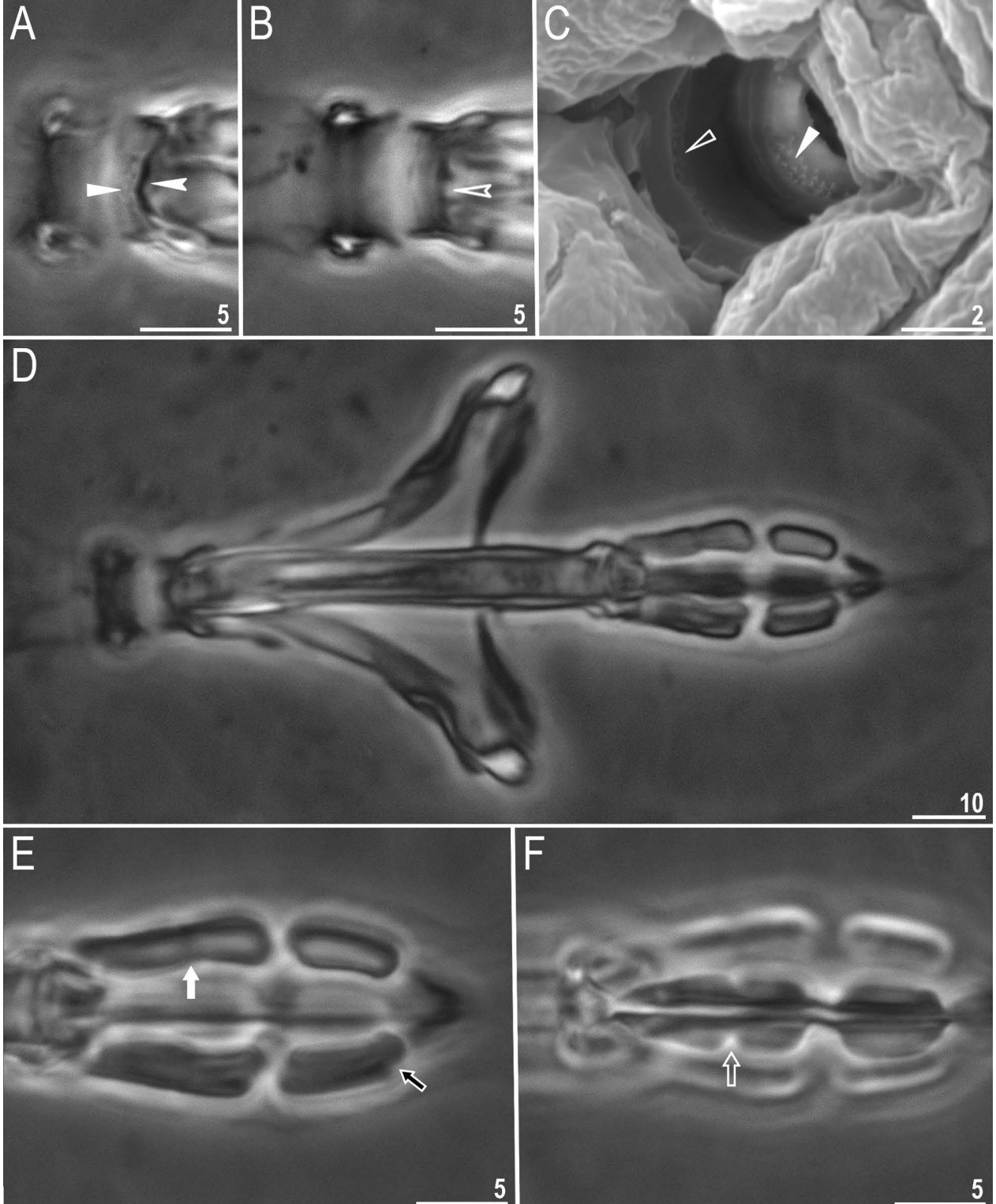

**Fig 2. *Macrobiotus sharopovi* sp. nov.: A – oral cavity armature, dorsal view (holotype, PCM); B – oral cavity armature, ventral view (holo-type, PCM); C – oral cavity armature (paratype, SEM); D – bucco-pharyngeal apparatus, dorsal view (holotype, PCM); E – placoid row, dorsal**

**view (holotype, PCM); F – placoid row, ventral view (holotype, PCM).** Empty unindented arrowhead represents first band of teeth, filled unindented arrowhead represents second band of teeth, filled indented arrowhead represents third band of teeth on the dorsal view, empty indented arrowhead represents third band of teeth on the ventral view, filled arrow represents first macroplacoid with central constriction on dorsal side, empty arrow represents first macroplacoid with central constriction on ventral side and filled black arrow represents second macroplacoid with sub-terminal constriction. Scale bars in µm.

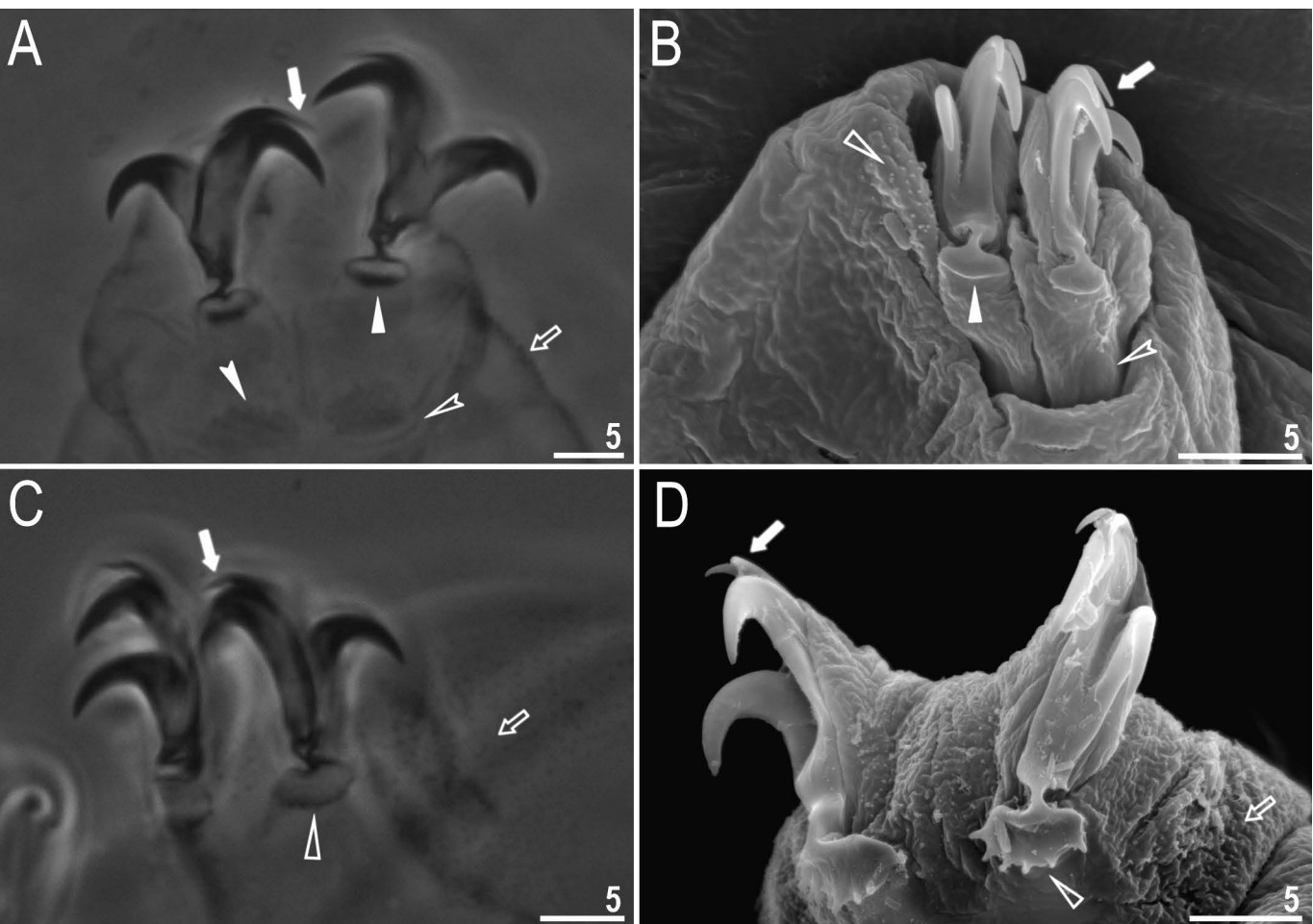

**Fig 3. *Macrobiotus sharopovi* sp. nov.: A – claws III (paratype, PCM); B – claws II (paratype, SEM); C – claws IV (paratype, PCM); D – claws IV (paratype, SEM).** Filled arrow represents accessory points on primary branches of claws, empty arrow represents granulations present on legs, filled unindented arrowhead represents smooth lunules, empty unindented arrowhead represents dentated lunules, filled indented arrowhead represents cuticular bar and empty indented arrowhead represents double muscle attachments. Scale bars in µm.

**Animals (measurements and statistics in** Table 1**).** Body white to slightly yellowish in living specimens and transparent after fixation in Hoyer's medium. Eyes present in all fixed specimens (Fig 1A). Entire cuticle covered with elliptical pores (0.8–1.8 µm in diameter) distributed throughout the body and clearly visible in both PCM and SEM (Fig 1B–D). The edges of cuticular pores, in the SEM, evidently thicker compared with surrounding cuticle. Tiny granules inside pores absent. Easily visible, continuous granulation on lateral side and above claws is present on legs I–IV (Fig 3A–D, empty arrow), with no patches. On the ventral side of legs granulation is absent.

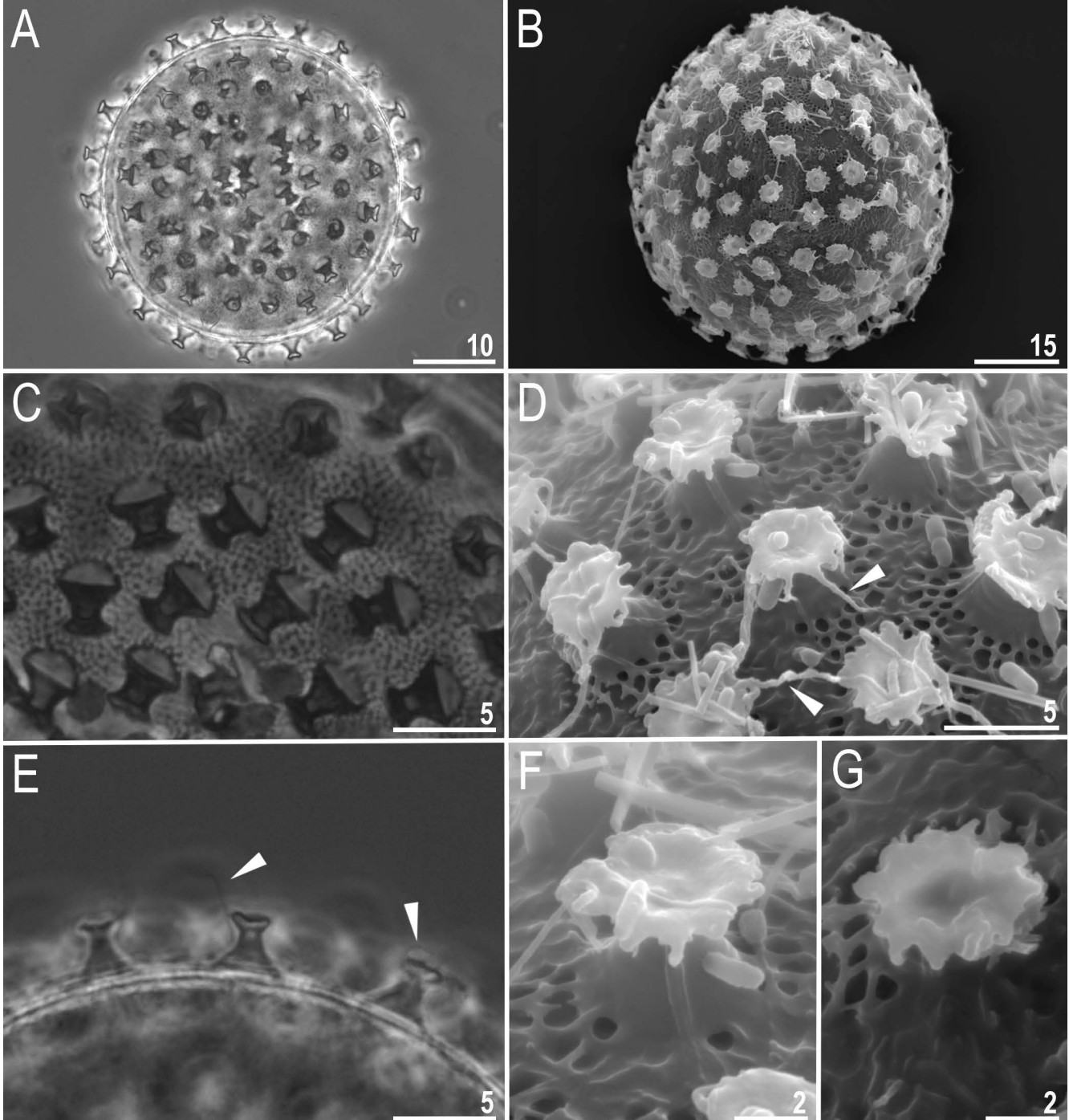

**Fig 4. *Macrobiotus sharopovi* sp. nov.: A–B – egg chorion (paratype, PCM and SEM, respectively); C–D – the surface between egg processes (paratype, PCM and SEM, respectively); E – egg circumference with egg processes midsections (paratype, PCM); F–G – egg processes (paratype, SEM).** Filled unindented arrowheads represent flexible filaments on the egg processes. Scale bars in μm.

Table 1. Measurements [in μm] and *pt* values of selected morphological structures of individuals of *Macrobiotus sharopovi* sp. nov., mounted in Hoyer's medium (N – number of specimens/structures measured; RANGE refers to the smallest and the largest structure among all measured specimens; SD – standard deviation, *pt* – ratio of the length of a given structure to the length of the buccal tube expressed as a percentage,? – lack of measurements due to unsuitable position of the structure).

| CHARACTER | N | RANGE μm | | | RANGE pt | | | MEAN μm | MEAN pt | SD μm | SD pt | Holotype μm | Holotype pt |
|---|---|---|---|---|---|---|---|---|---|---|---|---|---|
| Body length | 20 | 333 | – | 492 | | – | | 397 | | 45 | | 418 | |
| **Buccal tube** | | | | | | | | | | | | | |
| Buccal tube length | 20 | 29.7 | – | 38.7 | | – | | 34.0 | – | 2.1 | – | 34.0 | – |
| Stylet support insertion point | 20 | 21.6 | – | 28.4 | 70.1 | – | 75.3 | 25.1 | 73.7 | 1.5 | 1.3 | 25.6 | 75.3 |
| Buccal tube external width | 20 | 4.1 | – | 5.2 | 11.9 | – | 15.4 | 4.7 | 13.7 | 0.4 | 0.8 | 5.2 | 15.4 |
| Buccal tube internal width | 20 | 2.3 | – | 3.4 | 7.0 | – | 9.9 | 2.9 | 8.4 | 0.3 | 0.6 | 3.4 | 9.9 |
| Ventral lamina length | 15 | 17.0 | – | 23.0 | 57.3 | – | 62.1 | 20.5 | 59.8 | 1.5 | 1.5 | 20.8 | 61.1 |
| **Placoid lengths** | | | | | | | | | | | | | |
| Macroplacoid 1 | 20 | 8.0 | – | 11.3 | 25.3 | – | 31.2 | 9.2 | 27.0 | 0.9 | 1.5 | 9.6 | 28.3 |
| Macroplacoid 2 | 20 | 4.6 | – | 6.6 | 14.2 | – | 18.4 | 5.6 | 16.6 | 0.6 | 1.1 | 5.7 | 16.8 |
| Microplacoid | 20 | 2.0 | – | 3.0 | 6.0 | – | 8.7 | 2.5 | 7.3 | 0.2 | 0.6 | 3.0 | 8.7 |
| Macroplacoid row | 20 | 13.9 | – | 19.0 | 45.1 | – | 51.9 | 16.3 | 47.8 | 1.4 | 2.0 | 16.8 | 49.3 |
| Placoid row | 20 | 16.4 | – | 22.0 | 54.0 | – | 60.5 | 19.3 | 56.7 | 1.4 | 1.9 | 20.5 | 60.2 |
| **Claw I heights** | | | | | | | | | | | | | |
| External primary branch | 20 | 9.5 | – | 13.6 | 29.4 | – | 35.6 | 11.2 | 32.9 | 0.9 | 1.9 | 11.3 | 33.4 |
| External secondary branch | 20 | 7.2 | – | 10.1 | 22.0 | – | 28.7 | 8.5 | 24.9 | 0.8 | 1.9 | 9.8 | 28.7 |
| Internal primary branch | 20 | 8.8 | – | 12.5 | 27.5 | – | 33.1 | 10.3 | 30.1 | 0.8 | 1.6 | 10.5 | 30.9 |
| Internal secondary branch | 19 | 6.8 | – | 9.8 | 20.6 | – | 26.6 | 8.0 | 23.5 | 0.7 | 1.5 | 7.7 | 22.7 |
| **Claw II heights** | | | | | | | | | | | | | |
| External primary branch | 20 | 10.2 | – | 13.9 | 31.9 | – | 38.0 | 11.9 | 35.1 | 0.8 | 1.9 | 12.2 | 36.0 |
| External secondary branch | 20 | 7.9 | – | 10.4 | 24.0 | – | 29.7 | 9.2 | 27.1 | 0.7 | 1.8 | 10.0 | 29.5 |
| Internal primary branch | 20 | 8.7 | – | 13.0 | 28.4 | – | 35.6 | 11.0 | 32.3 | 0.9 | 2.0 | 11.2 | 32.9 |
| Internal secondary branch | 18 | 7.1 | – | 11.0 | 21.6 | – | 28.3 | 8.7 | 25.5 | 1.0 | 2.0 | 9.4 | 27.7 |
| **Claw III heights** | | | | | | | | | | | | | |
| External primary branch | 20 | 9.8 | – | 13.9 | 32.1 | – | 38.7 | 12.1 | 35.6 | 0.9 | 2.1 | 12.1 | 35.7 |
| External secondary branch | 18 | 7.5 | – | 12.3 | 24.2 | – | 31.9 | 9.2 | 27.2 | 1.1 | 2.2 | 8.5 | 24.9 |
| Internal primary branch | 20 | 9.3 | – | 13.3 | 29.3 | – | 36.4 | 11.1 | 32.7 | 0.8 | 2.0 | 10.8 | 31.8 |
| Internal secondary branch | 19 | 7.0 | – | 10.9 | 23.5 | – | 29.3 | 8.9 | 26.1 | 0.9 | 1.9 | 8.4 | 24.8 |
| **Claw IV lengths** | | | | | | | | | | | | | |
| Anterior primary branch | 18 | 9.8 | – | 15.4 | 31.6 | – | 39.8 | 12.2 | 35.6 | 1.2 | 2.3 | 12.6 | 37.1 |
| Anterior secondary branch | 17 | 7.7 | – | 11.0 | 23.2 | – | 31.0 | 9.4 | 27.6 | 1.0 | 2.3 | 9.8 | 28.9 |
| Posterior primary branch | 19 | 10.7 | – | 15.7 | 34.9 | – | 41.9 | 13.1 | 38.5 | 1.1 | 1.9 | 13.1 | 38.4 |
| Posterior secondary branch | 17 | 7.4 | – | 10.9 | 24.0 | – | 32.4 | 9.3 | 27.5 | 0.9 | 2.5 | ? | ? |

Bucco-pharyngeal apparatus of the *Macrobiotus* type, with ventral lamina and ten peribuccal lamellae. The OCA of the *patagonicus* type with second and third band of teeth visible with PCM (Fig 2A–B), however, with SEM the first band of teeth is visible as well (Fig 2C). The first band (visible only in SEM) consists of numerous extremely small teeth arranged in several rows situated anteriorly in the oral cavity behind the base of the peribuccal lamellae (Fig 2C, empty unindented arrowhead). The second band is situated between the ring fold and the third band of teeth and composed of numerous small cones arranged into 3–4 rows (Fig 2A–C, filled unindented arrowhead). The third band is located within the posterior portion of the oral cavity, between the second band of teeth and the buccal tube opening (Fig 2A–B). This band of teeth

**Table 2. Measurements [in µm] of selected morphological structures of eggs of *Macrobiotus sharopovi* sp. nov. mounted in Hoyer's medium (N – number of specimens/structures measured, RANGE refers to the smallest and the largest structure among all measured eggs; SD – standard deviation).**

| CHARACTER | N | RANGE | | | MEAN | SD |
|---|---|---|---|---|---|---|
| Egg bare diameter | 17 | 70.0 | – | 84.1 | 75.6 | 5.1 |
| Egg full diameter | 17 | 81.7 | – | 97.9 | 87.1 | 4.8 |
| Process height | 51 | 4.2 | – | 6.3 | 5.2 | 0.6 |
| Process base width | 51 | 4.4 | – | 6.8 | 5.6 | 0.6 |
| Process base/height ratio | 51 | 101% | – | 121% | 108% | 5% |
| Terminal disc width | 51 | 3.5 | – | 5.5 | 4.5 | 0.5 |
| Inter-process distance | 51 | 2.4 | – | 4.9 | 3.9 | 0.6 |
| Number of processes on the egg circumference | 15 | 20 | – | 24 | 21.6 | 1.5 |

is discontinuous and divided into a dorsal and ventral portion. Both dorsal and ventral teeth form a single transverse ridge (Fig 2A, filled indented arrowhead; Fig 2B, empty indented arrowhead).

Pharyngeal bulb spherical with triangular apophyses, two rod-shaped macroplacoids and a triangular microplacoid. Macroplacoid length sequence 2 < 1 (Fig 2D–F). The first macroplacoid with central constriction (Fig 2E–F, filled arrow and empty arrow). The second macroplacoid with sub-terminal constriction (Fig 2F, filled black arrow).

Claws Y-shaped of the *hufelandi* type, stout (Fig 3A–D). Under claws I–III, a divided cuticular bar (Fig 3A, filled indented arrowhead), and doubled muscle attachments (Fig 3A–B, empty indented arrowhead) are present and only faintly visible in PCM. Primary branches with distinct accessory points (Fig 3A–D, filled arrow) and a stalk connecting lunules and claws. Lunules under claws I–III smooth (Fig 3A–B, filled unindented arrowhead) and dentated under claws IV (Fig 3C–D, empty unindented arrowhead).

**Eggs (measurements and statistics in** Table 2**).** Spherical, white, ornamented and laid freely (Fig 4A–B) with *hufelandi* type egg chorion ornamentation (Fig 4C–D). The surface between processes reticulated with mesh of 0.6–1.1 µm in diameter. Egg processes in shape of concave cones with concave and indented smooth terminal discs and few long flexible smooth filaments on them (Fig 4D–G). Even though most of the filaments are broken and difficult to count, there are about 1–4 of them on the process's discs.

**Remarks.** Following normalization using Thorpe's method (see S1 Table), 14 of the 26 morphological traits examined showed statistical significance. Traits such as body length, macroplacoid 1, claw II internal primary branch, claw III external and internal secondary branch, claw IV anterior and posterior primary branch, claw IV anterior secondary branch, exhibited an allometric relationship characterized by a growth rate disproportionate to overall body size, as indicated by 'b' values significantly greater than 1. Conversely, the buccal tube external width, claw I external primary branch, claw external primary and secondary branches, claw III external and internal primary branch and claw IV posterior secondary branches displayed an allometric relationship with growth rates that are slower relative to body size, evidenced by 'b' values significantly less than 1.

**DNA sequences.** We obtained good quality sequences for the following molecular markers:

- 18S rRNA: GenBank: PV283176; 1260 bp long; (species voucher number: 1252.11).

- 28S rRNA: GenBank: PV283177; 777 bp long; (species voucher number: 1252.14).

- COI: GenBank: PV282532; 625 bp long; (species voucher number: 1252.12).

- ITS-2: GenBank: PV283179–80; 372–390 bp long; (species voucher numbers: 1252.11 and 1252.14).

**Genetic distances.** The ranges of uncorrected genetic p-distances between the molecular markers of *Mac. sharopovi* sp. nov. obtained in our study and the sequences of all species of the genus *Macrobiotus* available in GenBank are as follows (S5 Table):

- 18S rRNA: 0.2%–2.6% (1.6% on average), with the most similar being *Mac. shonaicus* Stec, Arakawa & Michalczyk 2018 (GenBank: MG757132) [61], and the least similar being *Mac. naginae* Vecchi, Stec, Vouri, Ryndov, Chartrain & Calhim 2022 (GenBank: OK663219–20) [62].

- 28S rRNA: 2.6%–10.1% (6.3% on average), with the most similar being *Mac. polypiformis* Roszkowka, Ostrowska, Stec, Janko & Kaczmarek 2017 (GenBank: KX810009) [40] and the least similar being *Mac. naginae* Vecchi, Stec, Vouri, Ryndov, Chartrain, Calhim & 2022 (GenBank: OK663230–31) [62].

- COI: 17.9%–28.03% (22.5% on average), with the most similar being *Mac. paulinae* Stec, Smolak, Kaczmarek & Michalczyk 2015 (GenBank: KT951668) [43], and the least similar being *Mac. rebecchi* Stec 2022 (GenBank: OP477442–43) [25].

- ITS2 rRNA: 15.9%–30.4% (23.6% on average), with the most similar being *Mac. papei* Stec, Kristensen & Michalczyk 2018 (GenBank: MH063921) [63] and the least similar being *Mac. vladimiri* Bertolani, Biserov, Rebecchi & Cesari 2011 [64] (GenBank: MN888347) [21].

**Morphological differential diagnosis.** The new species, based on the morphology of eggs, belongs to the *Mac. paulinae* morpho-group according to classification in Kaczmarek et al. [16] and it is most similar to *Mac. papei*, *Mac. paulinae*, *Mac. polypiformis* and *Mac. shonaicus*. But it differs specifically from the following:

1. ***Macrobiotus papei***, reported only from the type locality in Tanzania [63], by: absence of proximal and distal patches of granulation on the leg IV, larger pores/mesh on the egg chorion (0.6–1.1 µm in *Mac. sharopovi* sp. nov. *vs* 0.3–0.6 µm in *Mac. papei*), smaller number of filaments on the egg processes discs (1–4 in *Mac. sharopovi* sp. nov. *vs* 9–13 in *Mac. papei*) and smaller number of processes on the egg circumference (20–24 in *Mac. sharopovi* sp. nov. *vs* 26–32 in *Mac. papei*).

2. ***Macrobiotus paulinae***, reported only from the type locality in Kenya [34,43], by: the absence of dorso-lateral patches of granulation on the cuticle, larger diameter of cuticular pores (0.8–1.8 µm in *Mac. sharopovi* sp. nov. *vs* 0.3–0.5 µm in *Mac. paulinae*), presence of smooth filaments on the egg processes and larger pores/mesh on the egg chorion (0.6–1.1 µm in *Mac. sharopovi* sp. nov. *vs* 0.05–0.2 µm in *Mac. paulinae*).

3. ***Macrobiotus polypiformis***, reported only from the type locality in Ecuador [40], by: higher *pt* of the ventral lamina ($pt = 57.3$–$62.1$ in *Mac. sharopovi* sp. nov. *vs* $pt = 52.1$–$55.1$ in *Mac. polypiformis*), longer macroplacoid row and placoid row (13.9–19.0 µm [$pt = 45.1$–$51.9$] and 16.4–22.0 µm [$pt = 54.0$–$60.5$], respectively in *Mac. sharopovi* sp. nov. *vs* 9.0–11.8 µm [$pt = 34.3$–$39.9$] and 11.1–14.5 µm [$pt = 41.4$–$49.0$], respectively in *Mac. polypiformis*), presence of smooth filaments on egg processes discs and by smaller number of these filaments (1–4 in *Mac. sharopovi* sp. nov. *vs* 8–10 in *Mac. polypiformis*).

4. ***Macrobiotus shonaicus***, reported only from Japan [61,65], by: the absence of the cuticular bulge resembling pulvinus like structure and faint cuticular fold on legs I–III, larger diameter of cuticular pores (0.8–1.8 µm in *Mac. sharopovi* sp. nov. *vs* 0.2–0.4 µm in *Mac. shonaicus*), the presence of the reticulation on the egg surface between egg processes, smaller number of the egg processes on the egg circumference (20–24 in *Mac. sharopovi* sp. nov. *vs* 28–36 in *Mac. shonaicus*), presence of smooth filaments on egg processes discs and by smaller number of these filaments (1–4 in *Mac. sharopovi* sp. nov. *vs* 10–14 in *Mac. shonaicus*).

Genus: *Mesobiotus* (Vecchi, Cesari, Bertolani, Jönsson & Guidetti 2016) [11]
*Mesobiotus* cf. *coronatus* (de Barros 1942) [66]
(Figs 5–8, Tables 3–4)
Locality: Ecuador, Cotacachi-Cayapas National Park, Imbabura Province (00°17′31″N, 78°21′26″W; 3093 m asl), near Laguna de Cuicocha; moss on rock, 16th December 2014, leg. Milena Roszkowska, Pedro Rios Guayasamín and Łukasz Kaczmarek.

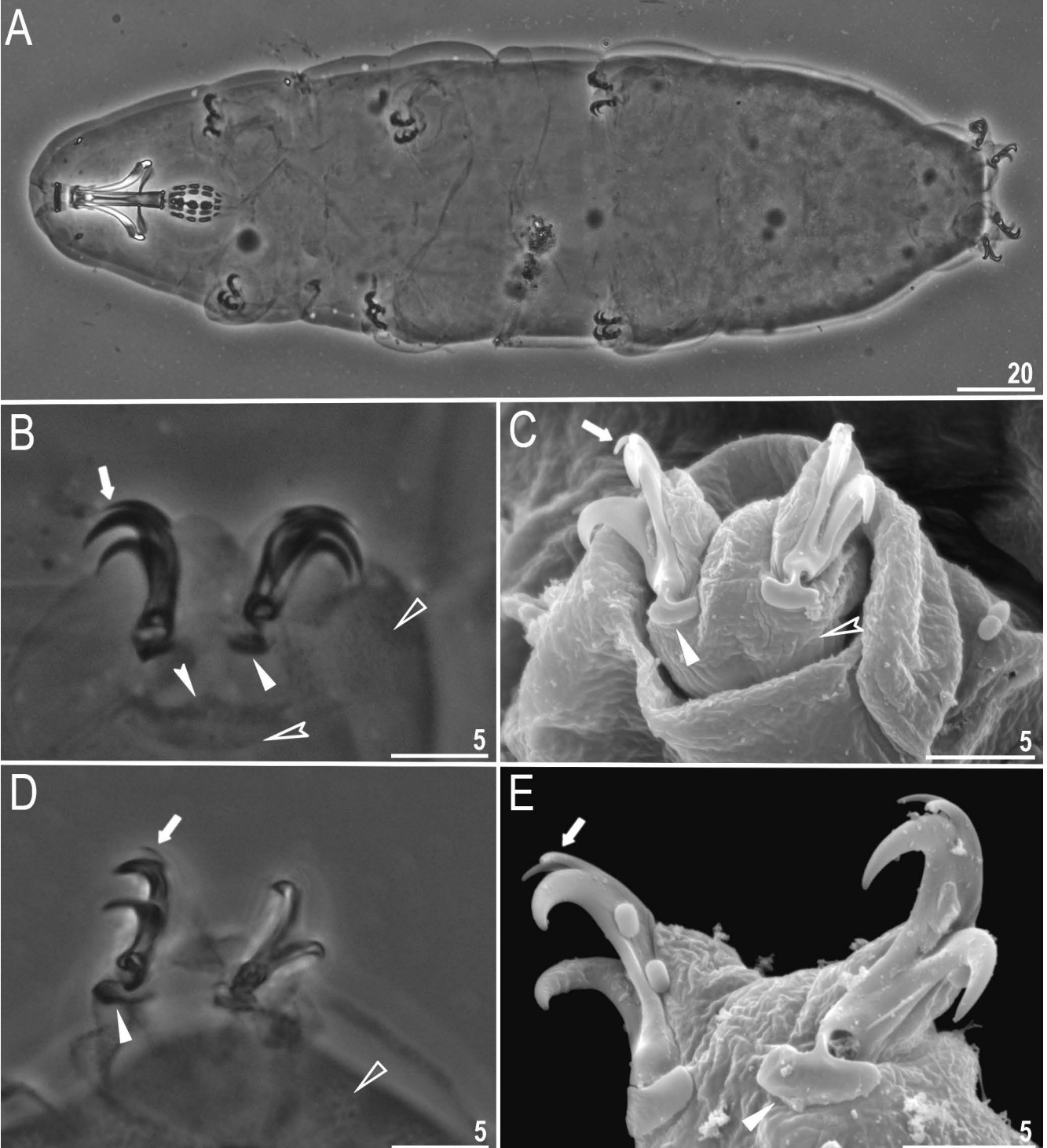

**Fig 5. *Mesobiotus* cf. *coronatus*: A – body, ventral view (PCM); B – claws III (PCM); C – claws II (SEM); D – claws IV (PCM); E – claws IV (SEM).** Filled arrow represents accessory points on primary branches of claws, filled unindented arrowhead represents smooth lunulas, empty unindented arrowhead represents granulations present on legs, filled indented arrowhead represents cuticular bar and empty indented arrowhead represents double muscle attachments. Scale bars in µm.

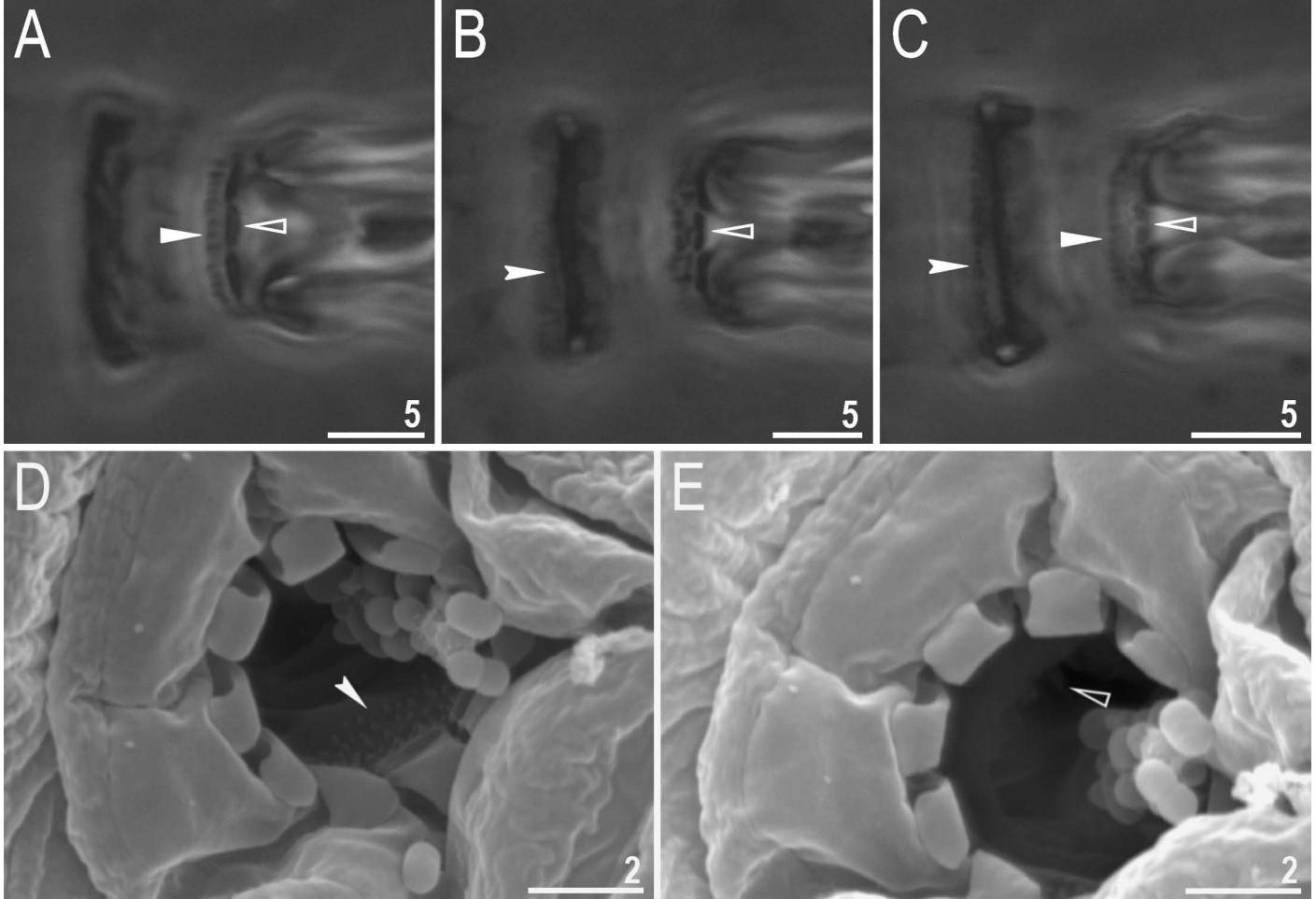

**Fig 6.** *Mesobiotus* cf. *coronatus*: A – oral cavity armature, dorsal view (PCM); B – oral cavity armature, ventral view (PCM); C – oral cavity armature, ventral view (PCM); D–E – oral cavity armature (SEM). Filled indented arrowhead represents the first band of teeth, filled unindented arrowhead represents the second band of teeth and empty unindented arrowhead represents the third band of teeth. Scale bars in µm.

Material examined: In total 119 specimens (79 animals and 40 eggs) were examined. The 47 animals and 30 eggs were mounted on microscope slides in Hoyer's medium, 25 animals and 10 eggs were fixed on SEM stubs and seven animals were processed for DNA sequencing, out of which we found good quality barcodes from one specimen.

Depositories: Examined material (47 animals and 30 eggs; slides: 1211-*, where the asterisk can be substituted by any of the following numbers: 1, 2, 3, 4, 5, 6, 10, 11, 12, 16, 17, 18 (animals) and 7, 8, 9, 13, 14, 15 (eggs)) are deposited in the Department of Animal Taxonomy and Ecology, Institute of Environmental Biology, Adam Mickiewicz University, Poznań, Uniwersytetu Poznańskiego 6, 61–614 Poznań, Poland.

**Animals (all measurements and statistics in** Table 3**).** Live specimens white, but after fixation in Hoyer's medium transparent. Eyes present in all fixed specimens (Fig 5A). Dorsal, lateral and ventral cuticle smooth and without pores. Continuous granulation on lateral side and above claws is present on legs I–IV and well visible in PCM (Fig 5B–E, empty unindented arrowhead); on the ventral side of legs granulation is absent. Thin singular cuticular bars (Fig 5B, filled indented arrowhead) and double muscle attachments under claws I-III are present (Fig 5B–C, empty indented arrowhead).

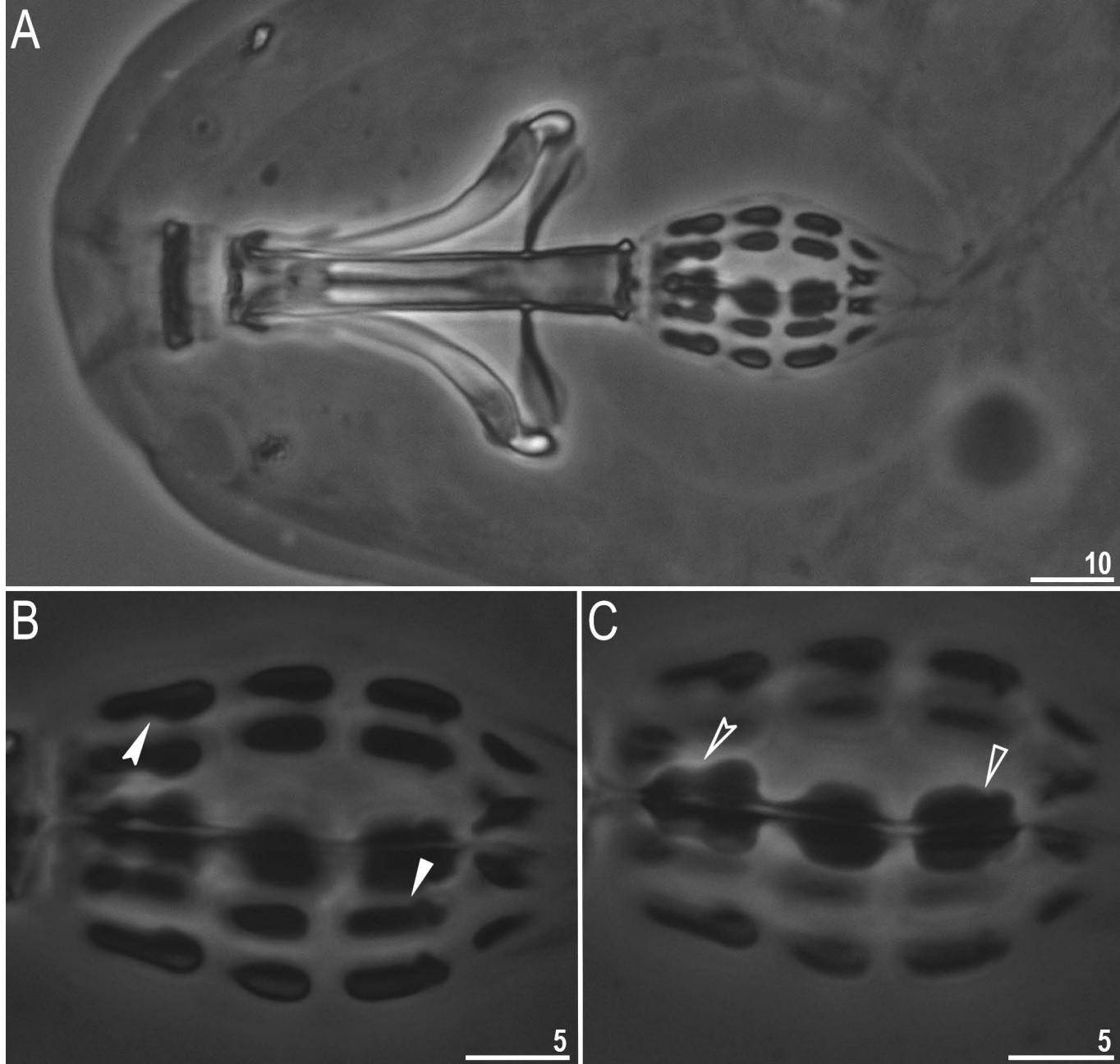

**Fig 7. *Mesobiotus* cf. *coronatus*: A – bucco-pharyngeal apparatus, dorsal view; B – placoid row, dorsal view; C – placoid row, ventral view.** Filled indented arrowhead and empty indented arrowhead represent central constriction in first macroplacoid on the dorsal side and on the ventral side respectively, filled unindented arrowhead and empty unindented arrowhead represent subterminal constriction on the third macroplacoid on the dorsal side and on the ventral side respectively. All PCM. Scale bars in µm.

Bucco-pharyngeal apparatus of *Mesobiotus* type with ten peribuccal lamellae and the ventral lamina. The OCA of the *harmsworthi* type with three bands of teeth, visible in PCM (Fig 6A–C). The first band is composed of small teeth that are clearly visible in PCM as granules (Fig 6D–E, filled indented arrowhead). The second band is composed of teeth that are arranged in a row parallel to the main axis of the buccal tube as comma-shaped ridges (Fig 6A, C, filled

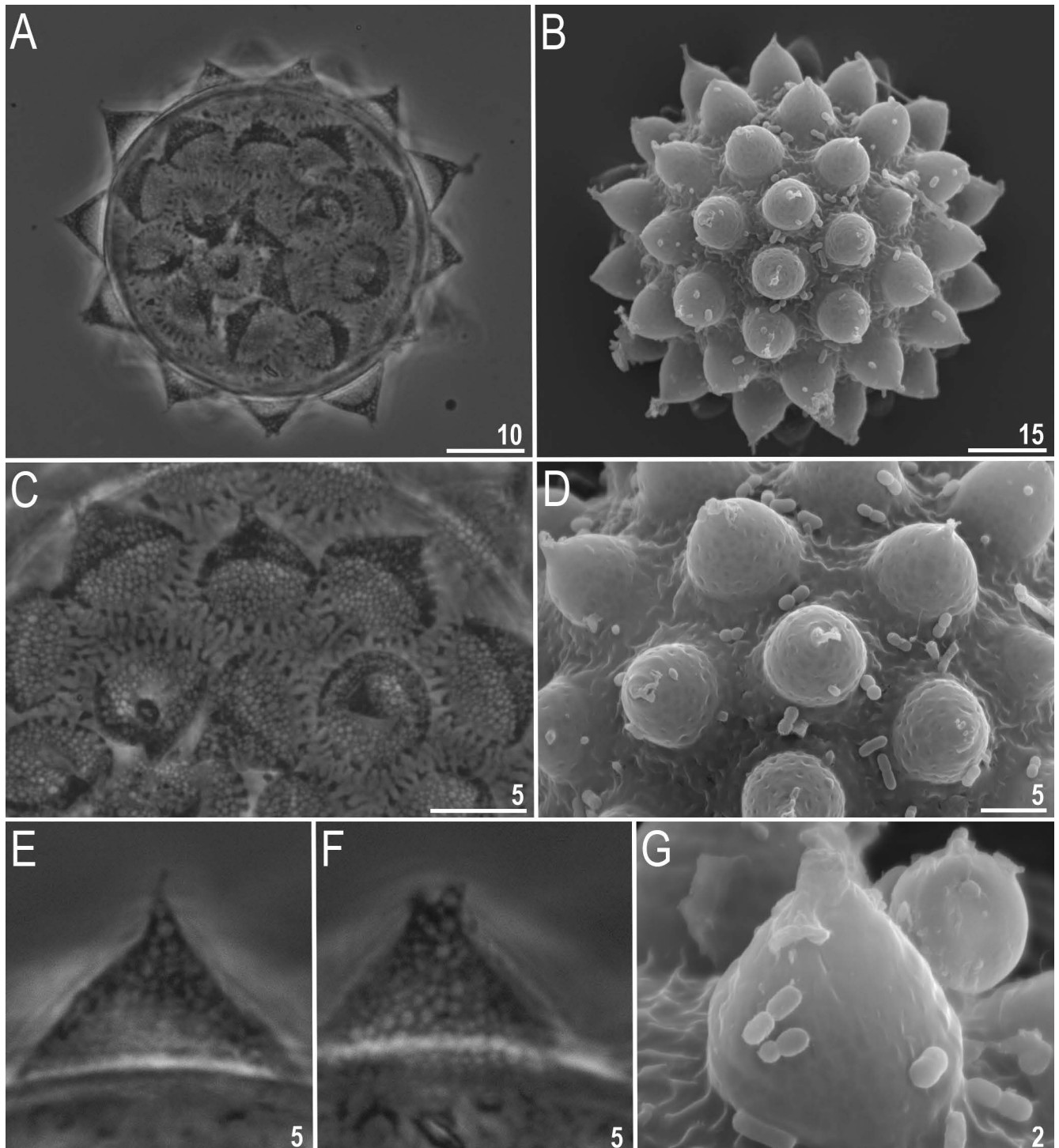

**Fig 8. *Mesobiotus* cf. *coronatus*: A–B – egg chorion (PCM and SEM, respectively); C–D – the surface between egg processes (PCM and SEM, respectively); E–F – egg processes midsections (PCM); G – egg processes (SEM).** Scale bars in μm.

Table 3. Measurements [in µm] and *pt* values of selected morphological structures of individuals of *Mesobiotus* cf. *coronatus*, mounted in Hoyer's medium (N – number of specimens/structures measured; RANGE refers to the smallest and the largest structure among all measured specimens; SD – standard deviation, *pt* – ratio of the length of a given structure to the length of the buccal tube expressed as a percentage,? – lack of measurements due to unsuitable position of the structure).

| CHARACTER | N | RANGE | | | | | | MEAN | | SD | |
|---|---|---|---|---|---|---|---|---|---|---|---|
| | | µm | | | *pt* | | | µm | *pt* | µm | *pt* |
| Body length | 20 | 329 | – | 459 | | – | | 392 | | 38 | |
| **Buccal tube** | | | | | | | | | | | |
| Buccal tube length | 20 | 30.6 | – | 39.7 | | – | | 36.6 | – | 2.5 | – |
| Stylet support insertion point | 20 | 23.1 | – | 30.9 | 74.5 | – | 78.7 | 28.0 | 76.6 | 2.0 | 1.1 |
| Buccal tube external width | 20 | 4.4 | – | 7.0 | 13.9 | – | 17.7 | 5.7 | 15.5 | 0.7 | 1.1 |
| Buccal tube internal width | 20 | 3.3 | – | 5.5 | 9.6 | – | 14.1 | 4.2 | 11.5 | 0.6 | 1.2 |
| Ventral lamina length | 17 | 18.1 | – | 24.6 | 58.1 | – | 62.9 | 22.3 | 61.0 | 1.7 | 1.3 |
| **Placoid lengths** | | | | | | | | | | | |
| Macroplacoid 1 | 20 | 4.5 | – | 7.1 | 13.6 | – | 17.9 | 5.7 | 15.6 | 0.7 | 1.3 |
| Macroplacoid 2 | 20 | 3.6 | – | 5.8 | 9.9 | – | 14.7 | 4.5 | 12.1 | 0.6 | 1.1 |
| Macroplacoid 3 | 20 | 4.3 | – | 6.9 | 13.0 | – | 17.3 | 5.3 | 14.5 | 0.7 | 1.2 |
| Microplacoid | 20 | 2.6 | – | 3.8 | 7.5 | – | 10.5 | 3.1 | 8.6 | 0.3 | 0.7 |
| Macroplacoid row | 20 | 14.4 | – | 22.6 | 46.0 | – | 57.1 | 18.1 | 49.3 | 2.2 | 3.3 |
| Placoid row | 20 | 17.8 | – | 28.0 | 57.2 | – | 70.8 | 22.3 | 60.9 | 2.5 | 3.5 |
| **Claw I heights** | | | | | | | | | | | |
| External primary branch | 18 | 8.8 | – | 11.5 | 25.4 | – | 30.7 | 10.2 | 27.9 | 0.9 | 1.5 |
| External secondary branch | 18 | 6.6 | – | 8.7 | 18.0 | – | 23.2 | 7.7 | 21.0 | 0.7 | 1.7 |
| Internal primary branch | 18 | 8.6 | – | 11.3 | 24.2 | – | 28.6 | 9.7 | 26.5 | 0.7 | 1.3 |
| Internal secondary branch | 18 | 5.9 | – | 8.0 | 16.4 | – | 22.8 | 7.1 | 19.4 | 0.7 | 1.5 |
| **Claw II heights** | | | | | | | | | | | |
| External primary branch | 20 | 9.2 | – | 11.9 | 26.9 | – | 31.4 | 10.6 | 29.1 | 0.7 | 1.3 |
| External secondary branch | 20 | 6.3 | – | 8.9 | 17.4 | – | 23.8 | 7.8 | 21.2 | 0.7 | 1.6 |
| Internal primary branch | 19 | 8.3 | – | 11.2 | 25.7 | – | 30.8 | 10.1 | 27.7 | 0.8 | 1.6 |
| Internal secondary branch | 17 | 6.4 | – | 8.6 | 17.1 | – | 24.0 | 7.7 | 21.0 | 0.7 | 1.7 |
| **Claw III heights** | | | | | | | | | | | |
| External primary branch | 19 | 9.1 | – | 11.9 | 26.6 | – | 32.3 | 10.7 | 29.3 | 0.8 | 1.7 |
| External secondary branch | 18 | 7.0 | – | 8.9 | 18.3 | – | 25.8 | 7.9 | 21.7 | 0.6 | 2.0 |
| Internal primary branch | 19 | 8.9 | – | 11.5 | 24.4 | – | 32.2 | 10.3 | 28.3 | 0.7 | 1.9 |
| Internal secondary branch | 18 | 6.2 | – | 8.6 | 18.0 | – | 23.9 | 7.5 | 20.6 | 0.6 | 1.8 |
| **Claw IV lengths** | | | | | | | | | | | |
| Anterior primary branch | 19 | 10.5 | – | 13.7 | 29.5 | – | 36.4 | 11.8 | 32.5 | 0.9 | 1.9 |
| Anterior secondary branch | 19 | 7.3 | – | 10.4 | 21.2 | – | 26.7 | 8.6 | 23.6 | 0.7 | 1.5 |
| Posterior primary branch | 20 | 10.8 | – | 14.3 | 30.6 | – | 38.4 | 12.4 | 34.0 | 1.0 | 2.1 |
| Posterior secondary branch | 20 | 7.6 | – | 10.2 | 21.3 | – | 26.3 | 8.8 | 24.0 | 0.7 | 1.6 |

unindented arrowhead). The third band is composed of teeth that are arranged as a system of three dorsal and three ventral transverse ridges (Fig 6A–C, E, empty unindented arrowhead). The central ventral ridge is bar-shaped, resembling a single tooth (Fig 6B; sometimes divided into two smaller teeth as on Fig 6C), while the lateral ventral ridges are curved, longer, and thicker than the middle one, but they are thinner at the external extremities (Fig 6B–C). Pharyngeal bulb round with apophyses, three rod-shaped macroplacoids and large microplacoid situated in close proximity to the

**Table 4. Measurements [in µm] of selected morphological structures of eggs of *Mesobiotus* cf. *coronatus,* mounted in Hoyer's medium (N – number of specimens/structures measured, RANGE refers to the smallest and the largest structure among all measured eggs; SD – standard deviation).**

| CHARACTER | N | RANGE | | | MEAN | SD |
|---|---|---|---|---|---|---|
| Egg bare diameter | 11 | 61.9 | – | 72.9 | 66.7 | 4.5 |
| Egg full diameter | 10 | 80.7 | – | 93.7 | 86.8 | 4.9 |
| Process height | 26 | 10.6 | – | 13.7 | 12.0 | 0.9 |
| Process base width | 26 | 10.1 | – | 14.0 | 12.2 | 1.1 |
| Process base/height ratio | 26 | 78% | – | 119% | 102% | 12% |
| Inter-process distance | 24 | 2.5 | – | 3.8 | 3.0 | 0.3 |
| Number of processes on the egg circumference | 4 | 11 | – | 14 | 12.5 | 1.3 |

final macroplacoid (Fig 7A). Macroplacoid lengths sequence is 2<3<1. Both the first and the second macroplacoids are widened at the end, and the first macroplacoid is constricted. (Fig 7B–C, filled indented arrowhead and empty indented arrowhead). The third macroplacoid lacks a terminal protrusion, but has a shallow, conspicuous subterminal constriction (Fig 7B–C, filled unindented arrowhead and empty unindented arrowhead).

Claws of the *Mesobiotus* type with a short common tract with septa at its proximal part (Fig 5B–E). All primary branches with accessory points (Fig 5B–E, filled arrow). External and posterior claws longer than internal and anterior claws. Claws on leg IV larger than claws on other legs. Lunules under all claws smooth (Fig 5B–E, filled unindented arrowhead).

**Eggs (all measurements and statistics in** Table 4**).** Spherical, white and laid freely (Fig 8A–B). Egg chorion with processes in the shape of wide sharp cones with thick, but flexible, short smooth endings. Egg surface between processes is sculptured (Fig 8C, in PCM), and in SEM (Fig 8D) this sculpture shown as wrinkles of diverse shapes. Egg processes bases are surrounded by a crown of slightly elongated thickenings (Fig 8C–D), and in SEM surface of processes is smooth or rarely with small pores on processes bases. The labyrinthine layer is visible under PCM as a reticulum in the processes' walls, with varying mesh size uniformly distributed within the process walls (Fig 8C, E–F). In SEM, instead, egg processes covered by shallow hollows, distributed regularly (Fig 8D, G). Apical parts of some processes are bifurcated or divided into short filaments (Fig 8E–G).

**Remarks.** Following normalization using Thorpe's method (see S2 Table), 22 of the 28 morphological traits examined showed statistical significance. Traits such as buccal tube external width, buccal tube internal width, and macroplacoids 1, 2, and 3, along with the macroplacoid row, exhibited an allometric relationship characterized by a growth rate disproportionate to overall body size, as indicated by 'b' values significantly greater than 1. Conversely, the microplacoid and the claws (including the external secondary and internal branches of claw I, as well as claws II, III, and IV) displayed an allometric relationship with growth rates that are slower relative to body size, evidenced by 'b' values significantly less than 1.

For the taxonomic remarks see the Conclusions section below

**DNA sequences.** We obtained good quality sequences for the following molecular markers:

- 28S rRNA: GenBank: PV283175; 746 bp long (species voucher number: 1211.5).

- COI: GenBank: PV282531; 624 bp long (species voucher number: 1211.5).

- ITS-2: GenBank: PV283178; 362 bp long (species voucher number: 1211.5).

**Genetic distances.** The ranges of uncorrected genetic p-distances between the molecular markers of *Meb.* cf. *coronatus* obtained in our study and the sequences of all species of the genus *Mesobiotus* available in GenBank are as follows (S6 Table):

- 28S rRNA: 6.1%–13.7% (8.5% on average), with the most similar being *Meb. vulpinus* Tumanov, Androsova, Gavrilenko & Kalimullin 2024 (GenBank: OR805140–41) [22], and the least similar being *Meb. dilimanensis* Itang, Stec, Mapalo, Mirano-Bascos & Michalczyk 2020 (GenBank: MN257049) [67].

- COI: 19.3%–27.6% (22.9% on average), with the most similar being *Meb. harmsworthi* Murray 1907 [68] (GenBank: MN313170) [69], and the least similar being *Meb.* cf. *furciger* (GenBank: MW727958) [24].

- ITS2 rRNA: 11.8%–43.4% (22.9% on average), with the most similar being *Meb. occultatus* Kaczmarek, Zawierucha, Buda, Stec, Gawlak, Michalczyk & Roszkowska 2018 (GenBank: MH197155) [69], and the least similar being *Meb. marmoreus* Stec 2021 (GenBank: OL257861–63) [70].

Superfamily: Hypsibioidea Pilato 1969 [71] (in Marley et al. [72])
Family: Ramazzottiidae Sands, McInnes, Marley, Goodall-Copestake, Convey & Linse 2008 [26]
Genus: *Ramazzottius* Binda & Pilato 1986 [27]
*Ramazzottius syraxi* sp. nov. Polishchuk, Kayastha, Warguła & Kaczmarek
urn:lsid:zoobank.org:act:6A0E8F04-89BD-4C61-B6BF-B319592BC858
(Figs 9–13, Tables 5,6)

Type locality: Ecuador, Cotacachi-Cayapas National Park, Imbabura Province (00°17'46''N, 78°20'52''W; 3122 m asl), near Laguna de Cuicocha; mosses and lichens on shrubs, 16th December 2014, leg. Łukasz Kaczmarek, Milena Roszkowska and Pedro Rios Guayasamín.

Additional locality: Ecuador, Cotacachi-Cayapas National Park, Imbabura Province (00°18'02''N, 78°20'47''W; 3176 m asl), near Laguna de Cuicocha, cryptograms on shrubs, 16th December 2014, leg. Łukasz Kaczmarek, Milena Roszkowska and Pedro Rios Guayasamín.

Material examined: In total 13 specimens were examined (seven specimens from the type locality and six specimens from the additional locality) and mounted on microscope slides in Hoyer's medium.

Type depositories: The holotype (1219-13) and four paratypes (slides: 1219-*, where the asterisk can be substituted by any of the following numbers: 9, 22), as well as three microscope slides with six specimens from the additional locality (1132–2, 1132–3, 1132–4) are deposited in the Department of Animal Taxonomy and Ecology, Institute of Environmental Biology, Adam Mickiewicz University, Poznań, Uniwersytetu Poznańskiego 6, 61–614 Poznań, Poland. Two paratypes from the type locality (slide 1219–4) are deposited in Institute of Systematics and Evolution of Animals, Polish Academy of Sciences, Sławkowska 17, 31–016, Kraków, Poland.

Etymology: The authors dedicated the species name to the fantasy world "A Song of Ice and Fire" by George R.R. Martin. The species is named after the dragon Syrax belonged to the book character Rhaenyra Targaryen (from the book "Fire and Blood") due to the visual similarity caused by the cuticle appearance of *Ram. syraxi* sp. nov.

**Animals (all measurements and statistics in Tables 5,6).** Body colour yellow-orange in living specimens. Eyes absent in live animals. A pair of elliptical organs present on the head. The cuticle of the dorso-lateral part of the body exhibits eight rows of transverse sculptured bands, which contain either several weakly developed gibbosities and a sculptured cuticle between them (concave areas) or a single long gibbosite (configuration: VIII:1-2-3-4-5-4-3-1) (Fig 9A–B), while the ventral cuticle is smooth, as well as cuticle on all legs (Figs 9C, 11A). Gibbosities are covered by pebble-shaped polygonal granules with ragged edges of various sizes (1.6–3.8 μm in diameter), and some kind of 'reticulum' is visible between the granules (Fig 10A–B, filled unindented arrowhead), which forms a polygonal pattern, but Fig 10C (filled indented arrowhead) shows that the granules are located separately from each other. However, we cannot confirm the absence of 'reticulum' without SEM microphotographs, and a visual analysis of a larger number of individuals is required. The less strongly sculptured concave areas are present in rows II, IV and VI between laterally situated gibbosities (Figs 9A–B, 10A), and these areas in turn covered by irregular shaped structures composed of small granules, and their size also varies (0.8–3.3 μm in diameter) (Fig 10D). In rows I and VIII, a wide and convex sculptured stripes are

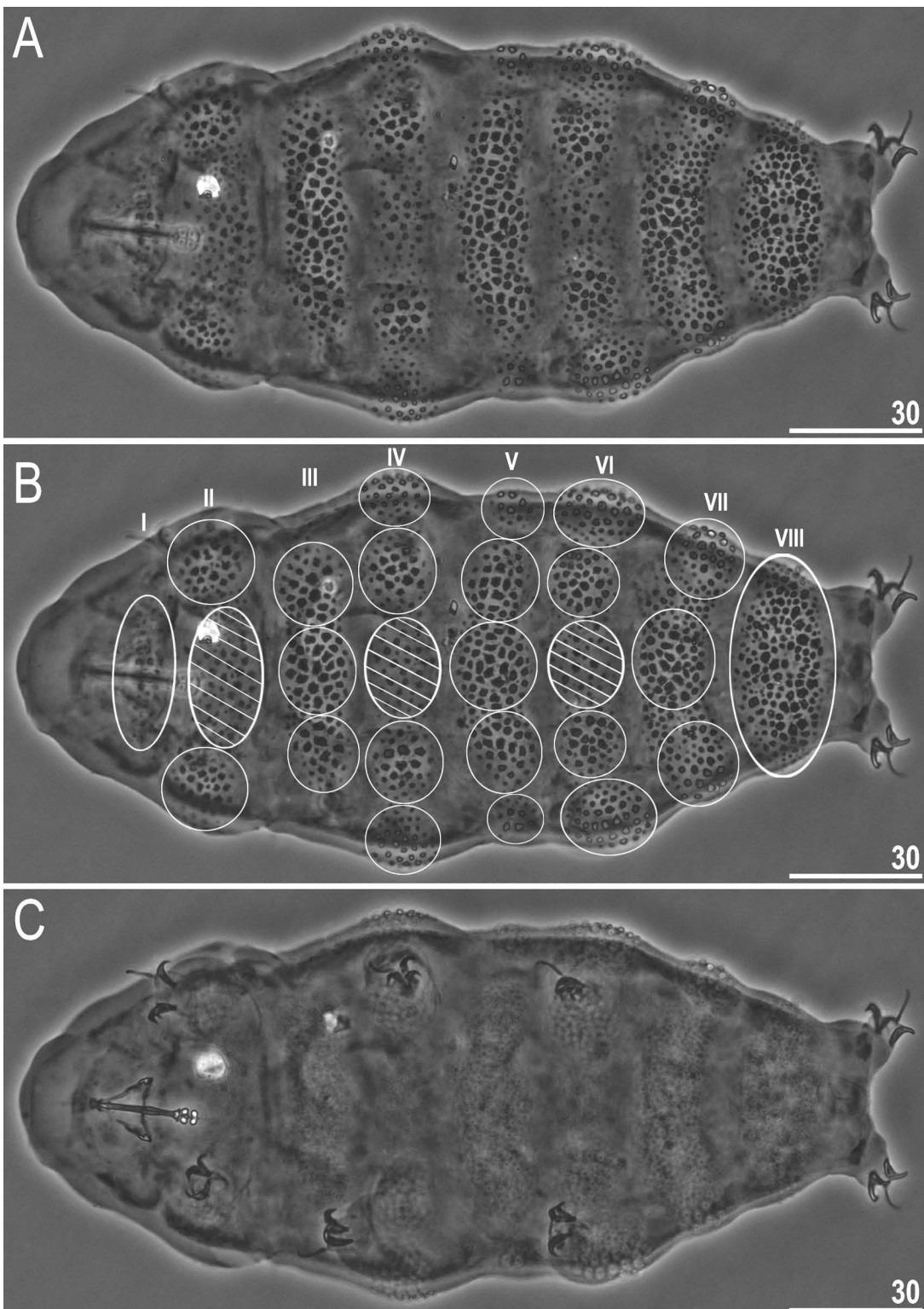

**Fig 9. _Ramazzottius syraxi_ sp. nov.: A – body, dorsal view (holotype); B – body, dorsal view (holotype) with numbering of eight rows of transverse sculptured bands (Roman numerals I–VIII), schematic marking of gibbosities on each band (white ellipses and circles) and concave areas between gibbosities (white striped ellipses); C – body, ventral view (holotype).** All PCM. Scale bars in μm.

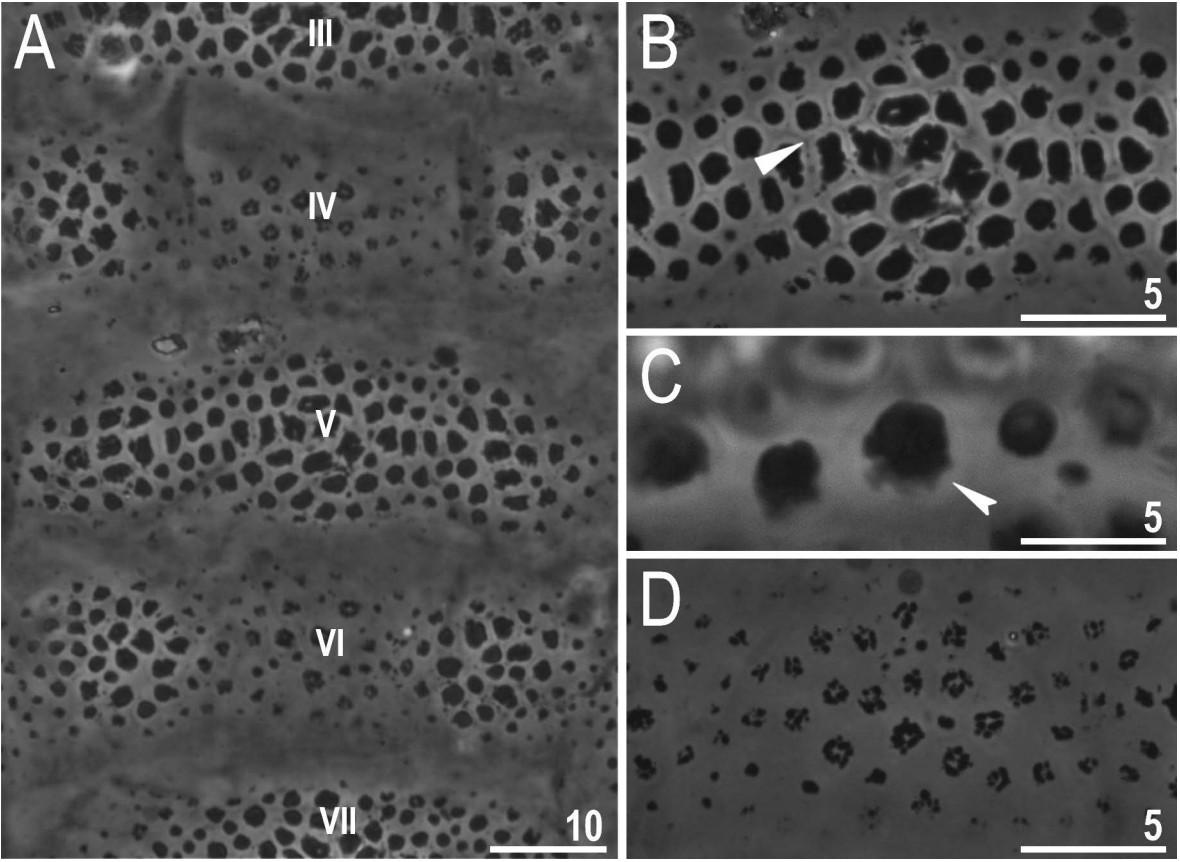

**Fig 10.** *Ramazzottius syraxi* sp. nov.: A – cuticle, dorsal view (holotype) with numbering of transverse sculptured bands (rows III–VII); B – gibbosite with granules, dorsal view (holotype), row V; C – a detailed view at the granules, dorso-lateral view (holotype); D – structures on concave area, dorsal view (holotype), row IV. Filled unindented arrowhead represents some kind of 'reticulum' between the granules, which forms a polygonal pattern, filled indented arrowhead represents the base of the granule on gibbosite. All PCM. Scale bars in µm.

present which are considered as single gibbosities (Fig 9A–B). The appearance of granules on gibbosities, and concave areas may vary slightly in different individuals (Fig 11A–D). Also, some specimens may have a small assemblance of granules on the dorsal side of the fourth pair of legs, near row VIII (Fig 11C, filled unindented arrowheads).

Peribuccal lamellae and peribuccal papulae are absent. The oral cavity armature not visible in PCM. Buccal-pharyngeal apparatus of the *Ramazzottius* type with well visible thickening just below a stylet insertion point (Fig 12A, filled unindented arrowhead). The pharyngeal bulb spherical with triangular apophyses and two macroplacoids (Fig 12B–C). Microplacoid and septulum are absent. Macroplacoids roundish and without constrictions (Fig 12B–C).

Claws of the *Ramazzottius* type (Fig 13A–B), i.e., two claws of the same leg clearly different from each other in size and shape. Primary branches of external claws and posterior claws longer than the primary branches of internal claws and anterior claws. The bases of all claws have a smooth pseudolunules (Fig 13A–B, filled unindented arrowhead). Primary branches of external/posterior claws with cuticular flexible portions, connected to the secondary branches. Accessory points present on all primary branches (Fig 13A–B, filled arrow).

Eggs unknown.

**Remarks.** There have been efforts to obtain DNA sequences for *Ram. syraxi* sp. nov. Due to the small number of specimens, only two individuals were used for DNA sequencing to analyse the markers 18S, 28S, COI and ITS-2. In both

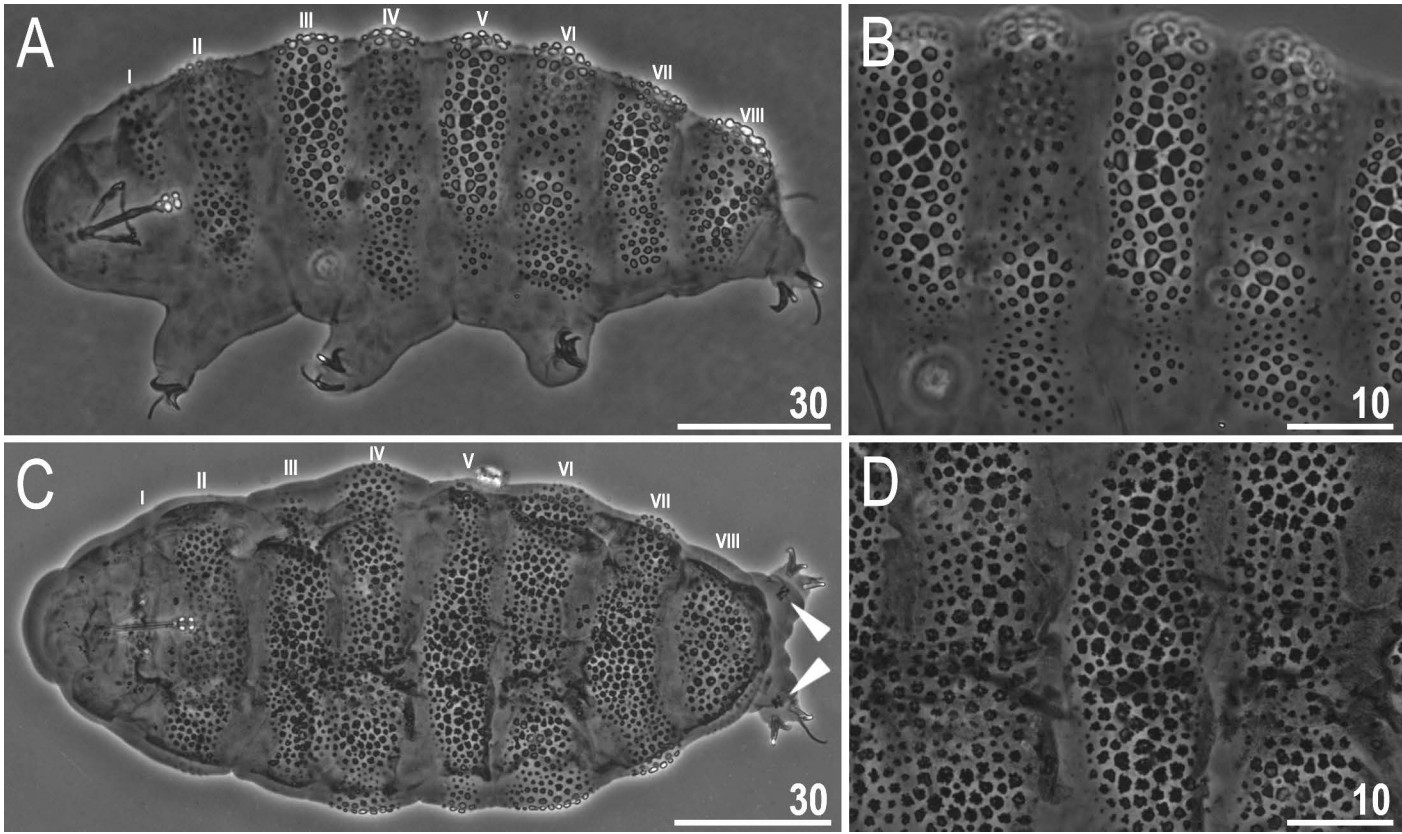

**Fig 11.** *Ramazzottius syraxi* **sp. nov.: A – body, dorso-lateral view (paratype, type locality) with numbering of rows of transverse sculptured bands (Roman numerals I–VIII); B – cuticle of the specimen from the Fig 11A, dorso-lateral view (paratype, type locality); C – body, dorsal view (paratype, type locality) with numbering of rows of transverse sculptured bands (Roman numerals I–VIII); D – cuticle of the specimen from the Fig 11C, dorsal view (paratype, type locality).** Two filled unindented arrowheads represent a small assemblance of granules on the dorsal side of the fourth pair of legs. All PCM. Scale bars in µm.

cases, the sequences were contaminated or failed to produce reliable results. Therefore, we decided not to publish poor quality genetic data and focus solely on the morphological description.

**Morphological differential diagnosis.** *Ramazzottius syraxi* sp. nov. is characterized by the presence of gibbosities and well visible sculpture on dorsal side of the body and it is similar to *Ram. baumanni*, *Ram.* cf. *baumanni* and *Ram.* aff. *baumanni* sp. can., *Ram. belubellus* Bartels, Nelson, Kaczmarek & Michalczyk 2011 [73], *Ram. saltensis* (Claps and Rossi 1984) [74] and *Ram. szeptycki* (Dastych 1980) [75]. However, new species differs specifically from:

1. ***Ramazzottius baumanni***, reported from Chile (type locality) [76], Argentina, Uruguay (see review in Kaczmarek et al. [32]), Canada and USA (see review in Kaczmarek et al. [33]), New Zealand [77] and Japan [78,79], by: different number of rows of transverse sculptured bands (eight rows in *Ram. syraxi* sp. nov. *vs* nine rows in *Ram. baumanni*), small gibbosities and concave areas between them within the rows II–VII (large elongated gibbosities, in particular on rows IV–VIII in *Ram. baumanni*), the lack of granulation on the head and smooth dorsal side of the cuticle on legs or just with the small assemblance of granules on the fourth pair (strong granulation on the dorsal side of all legs in *Ram. baumanni*, sometimes with a gibbosities on the fourth pair).

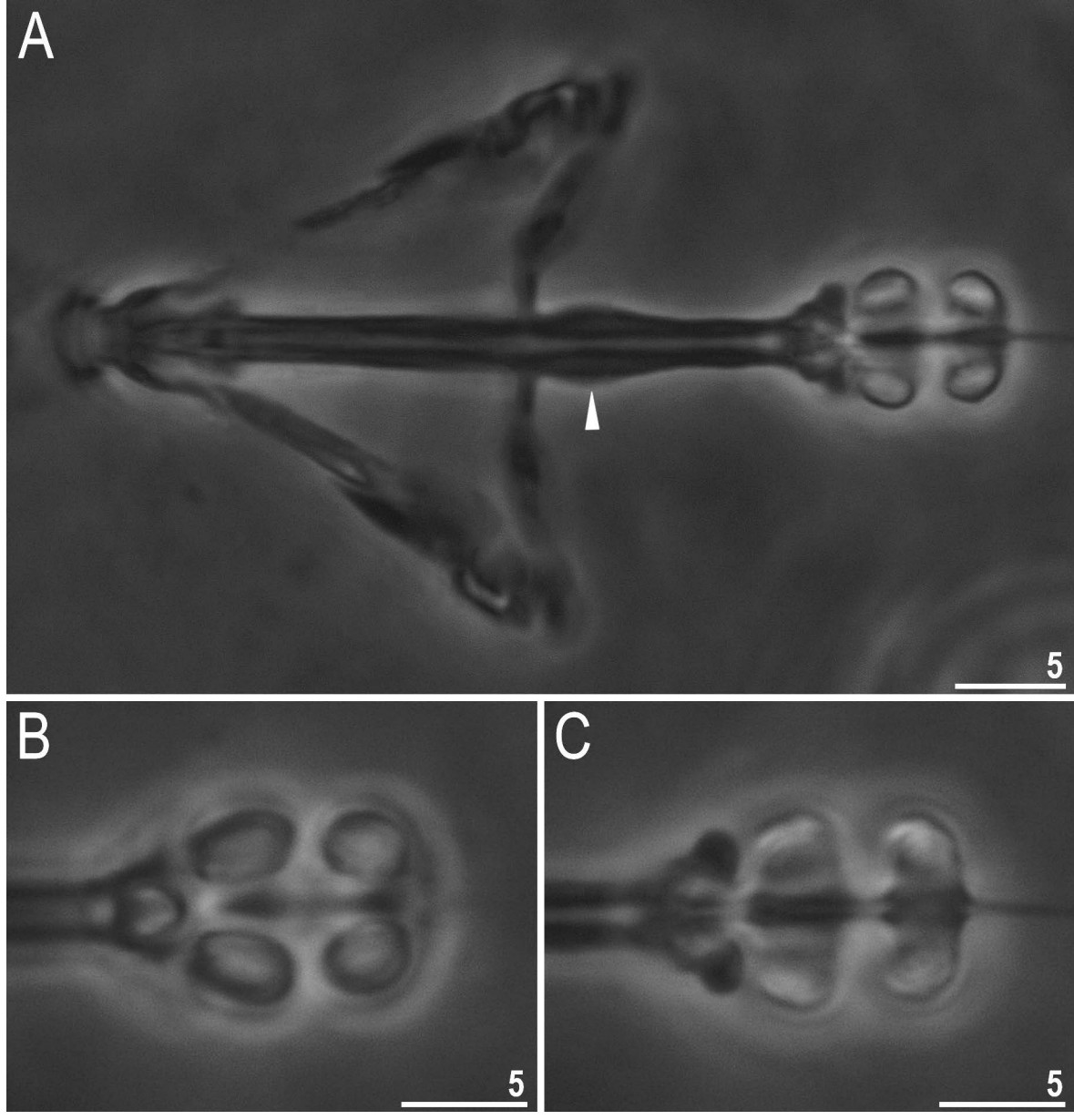

**Fig 12.** *Ramazzottius syraxi* **sp. nov.: A – bucco-pharyngeal apparatus, dorsal view (holotype); B – placoid row, dorsal view (holotype); C – placoid row, ventral view (holotype).** Filled unindented arrowhead represents thickening on buccal tube below a stylet insertion point. All PCM. Scale bars in μm.

2. *Ramazzottius* **cf.** *baumanni*, reported from Argentina [36], by: the lack of granulation on the head, smooth cuticle on the fourth pair of legs or just with the small assemblance of granules on the dorsal side (well-visible granulation on the fourth pair in *Ram.* cf. *baumanni*).

3. *Ramazzottius* **aff.** *baumanni* **sp. can.**, reported from Argentina [36], by: different number of rows of transverse sculptured bands (eight rows in *Ram. syraxi* sp. nov. *vs* seven rows in *Ram.* aff. *baumanni* sp. can.), presence of gibbosities

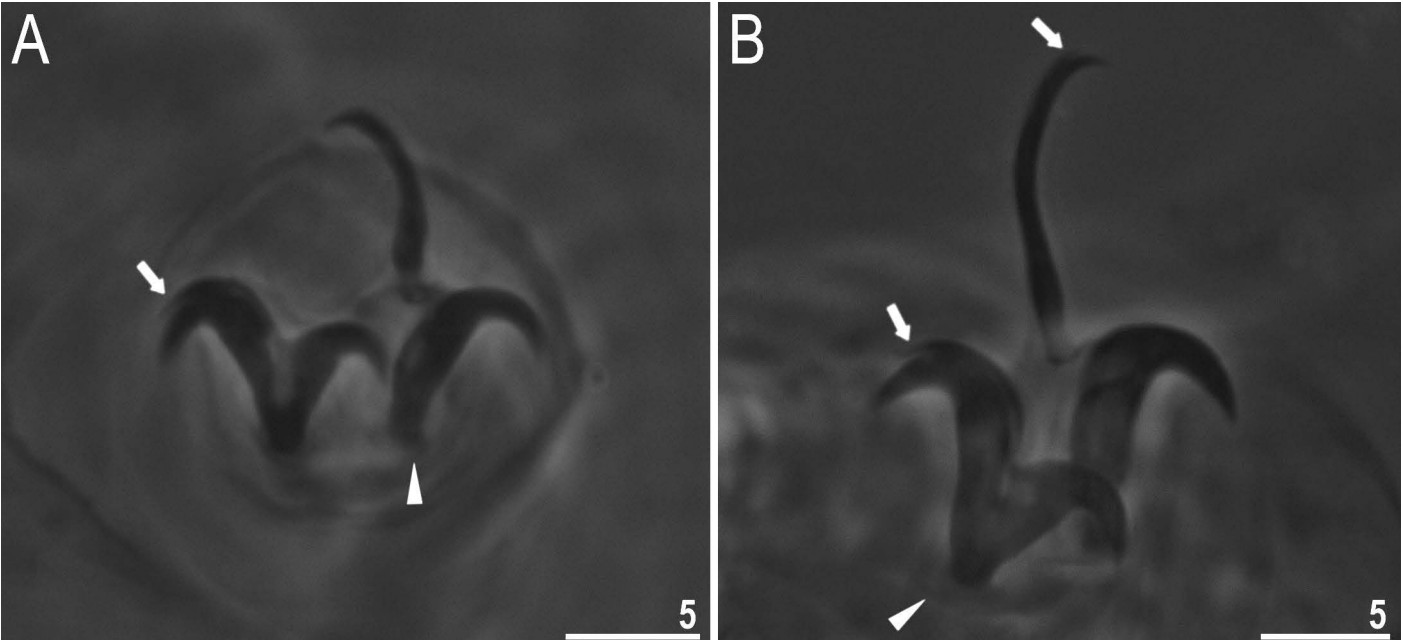

**Fig 13.** *Ramazzottius syraxi* sp. nov.: **A – claws II (paratype); B – claws IV (paratype).** Filled unindented arrowhead represents a smooth pseudolunule and filled arrow represents accessory points on primary branches of claws and. All PCM. Scale bars in μm.

on dorsal side of the body (no undulations or gibbosities on dorsal side of the body in *Ram*. aff. *baumanni* sp. can.) and larger and less numerous dorsal granules.

4. *Ramazzottius belubellus*, reported only from the type locality in USA [73], by: different dorsal sculpture (smaller and larger polygonal granules assembled in eight rows of transverse sculptured bands in *Ram. syraxi* sp. nov. *vs* sharp triangular, bulbous spines covering entire dorsum in *Ram. belubellus*), smooth cuticle on legs or just with the small assemblance of granules on the dorsal side of the fourth pair (small tubercles on external parts all legs in *Ram. belubellus*), presence of smooth pseudolunules on the bases of all claws (lunules and other cuticular thickenings on legs absent in *Ram. belubellus*).

5. *Ramazzottius saltensis*, known only from the type locality in Argentina [74] by: the different gibbosities configuration (VIII: 1-2-3-4-5-4-3-1 in *Ram. syraxi* sp. nov. *vs* VIII: 3-5-5-5-5-5-3-2 in *Ram. saltensis*) and different dorsal sculpture (smaller and larger polygonal granules in *Ram. syraxi* sp. nov. *vs* large, bubble-like granulation in *Ram. saltensis*).

6. *Ramazzottius szeptycki*, known only from Republic of South Africa (type locality) and Tanzania [75,80–83] by: different number of rows of transverse sculptured bands (eight rows in *Ram. syraxi* sp. nov. *vs* seven rows in *Ram. szeptycki*), different gibbosities configuration (VIII: 1-2-3-4-5-4-3-1 in *Ram. syraxi vs* VII: 8-8-10-10-10-10-4 in *Ram. szeptycki*) and different dorsal sculpture (smaller and larger polygonal granules in *Ram. syraxi vs* polygonal reticulation in *Ram. szeptycki*).

## Conclusions

In mainland Ecuador, 27 species of tardigrades have been reported so far [32,39–42]. Of this number, only three species of the genus *Macrobiotus* were reported, i.e., *Mac. hibiscus* De Barros 1942 [65], *Mac. polypiformis* and *Mac. dulciporus*

**Table 5. Measurements [in μm] and *pt* values of selected morphological structures of individuals of *Ramazzottius syraxi* sp. nov. from the type locality, mounted in Hoyer's medium (N – number of specimens/structures measured; RANGE refers to the smallest and the largest structure among all measured specimens; SD – standard deviation, *pt* – ratio of the length of a given structure to the length of the buccal tube expressed as a percentage,? – lack of measurements due to unsuitable position of the structure).**

| CHARACTER | N | RANGE | | | | | | MEAN | | SD | | Holotype | |
|---|---|---|---|---|---|---|---|---|---|---|---|---|---|
| | | μm | | | pt | | | μm | pt | μm | pt | μm | pt |
| Body length | 5 | 201 | – | 416 | | – | | 323 | | 91 | | 349 | |
| **Buccal tube** | | | | | | | | | | | | | |
| Buccal tube length | 5 | 25.1 | – | 36.5 | | – | | 31.8 | – | 4.4 | – | 31.6 | – |
| Stylet support insertion point | 5 | 14.3 | – | 20.7 | 56.6 | – | 59.4 | 18.5 | 58.3 | 2.6 | 1.4 | 18.8 | 59.4 |
| Buccal tube external width | 5 | 2.1 | – | 2.8 | 7.8 | – | 8.6 | 2.6 | 8.2 | 0.3 | 0.4 | 2.7 | 8.5 |
| Buccal tube internal width | 5 | 0.8 | – | 1.2 | 2.6 | – | 3.8 | 1.0 | 3.3 | 0.2 | 0.4 | 0.8 | 2.6 |
| **Placoid lengths** | | | | | | | | | | | | | |
| Macroplacoid 1 | 5 | 2.7 | – | 3.5 | 8.4 | – | 10.9 | 3.1 | 9.8 | 0.4 | 0.9 | 2.7 | 8.4 |
| Macroplacoid 2 | 5 | 2.3 | – | 3.2 | 8.3 | – | 9.2 | 2.8 | 8.7 | 0.4 | 0.4 | 2.7 | 8.6 |
| Macroplacoid row | 5 | 5.4 | – | 7.9 | 20.4 | – | 22.5 | 6.8 | 21.3 | 1.0 | 0.8 | 6.5 | 20.4 |
| **Claw I heights** | | | | | | | | | | | | | |
| External base | 5 | 4.2 | – | 10.2 | 16.9 | – | 28.9 | 8.0 | 24.7 | 2.3 | 4.6 | 8.4 | 26.7 |
| External primary branch | 5 | 7.3 | – | 12.7 | 28.9 | – | 36.1 | 10.6 | 33.2 | 2.1 | 2.6 | 10.8 | 34.0 |
| External secondary branch | 4 | 5.7 | – | 8.9 | 18.7 | – | 24.5 | 7.3 | 21.6 | 1.3 | 2.5 | 7.1 | 22.6 |
| Internal base | 5 | 4.3 | – | 8.2 | 16.7 | – | 22.4 | 6.3 | 19.5 | 1.6 | 2.6 | 5.3 | 16.7 |
| Internal primary branch | 4 | 5.6 | – | 9.3 | 21.0 | – | 26.5 | 7.1 | 23.1 | 1.6 | 2.4 | 6.6 | 21.0 |
| Internal secondary branch | 5 | 5.1 | – | 7.6 | 16.3 | – | 21.7 | 6.2 | 19.5 | 0.9 | 2.1 | 6.5 | 20.6 |
| **Claw II heights** | | | | | | | | | | | | | |
| External base | 5 | 7.2 | – | 10.8 | 28.7 | – | 30.7 | 9.4 | 29.6 | 1.5 | 0.8 | 9.5 | 30.1 |
| External primary branch | 5 | 8.2 | – | 13.2 | 32.5 | – | 36.2 | 11.0 | 34.3 | 2.0 | 1.7 | 11.2 | 35.5 |
| External secondary branch | 5 | 5.5 | – | 9.4 | 18.1 | – | 27.4 | 7.7 | 23.9 | 2.0 | 4.0 | 8.7 | 27.4 |
| Internal base | 5 | 5.1 | – | 8.4 | 19.9 | – | 23.8 | 6.8 | 21.3 | 1.4 | 1.7 | 6.3 | 20.0 |
| Internal primary branch | 3 | 6.0 | – | 9.5 | 23.7 | – | 27.2 | 7.7 | 24.9 | 1.8 | 1.9 | 7.5 | 23.7 |
| Internal secondary branch | 5 | 4.5 | – | 8.1 | 18.1 | – | 23.1 | 6.7 | 21.0 | 1.4 | 2.0 | 7.1 | 22.3 |
| **Claw III heights** | | | | | | | | | | | | | |
| External base | 5 | 7.1 | – | 10.7 | 24.0 | – | 30.5 | 8.7 | 27.4 | 1.4 | 2.6 | 9.1 | 28.8 |
| External primary branch | 5 | 8.1 | – | 14.0 | 29.5 | – | 40.0 | 11.2 | 34.9 | 2.5 | 4.1 | 11.9 | 37.7 |
| External secondary branch | 5 | 5.2 | – | 9.8 | 20.6 | – | 27.9 | 7.6 | 23.8 | 1.8 | 2.9 | 8.0 | 25.4 |
| Internal base | 5 | 4.9 | – | 7.8 | 17.3 | – | 22.6 | 6.5 | 20.2 | 1.3 | 2.2 | 7.2 | 22.6 |
| Internal primary branch | 2 | 5.6 | – | 8.7 | 22.2 | – | 24.8 | 7.1 | 23.5 | 2.2 | 1.9 | ? | ? |
| Internal secondary branch | 5 | 4.7 | – | 8.0 | 15.4 | – | 22.9 | 6.2 | 19.4 | 1.3 | 2.9 | 5.9 | 18.7 |
| **Claw IV lengths** | | | | | | | | | | | | | |
| Anterior base | 5 | 4.1 | – | 6.8 | 16.2 | – | 20.2 | 5.9 | 18.4 | 1.1 | 1.6 | 6.4 | 20.2 |
| Anterior primary branch | 5 | 5.8 | – | 9.0 | 22.9 | – | 24.8 | 7.5 | 23.7 | 1.3 | 0.8 | 7.3 | 22.9 |
| Anterior secondary branch | 4 | 4.0 | – | 7.6 | 15.8 | – | 21.7 | 6.1 | 19.0 | 1.5 | 3.1 | 6.9 | 21.7 |
| Posterior base | 5 | 6.1 | – | 10.5 | 22.9 | – | 29.9 | 8.4 | 26.3 | 1.5 | 2.8 | 8.4 | 26.5 |
| Posterior primary branch | 5 | 8.7 | – | 17.3 | 34.8 | – | 49.3 | 13.9 | 43.2 | 3.5 | 5.5 | 13.7 | 43.2 |
| Posterior secondary branch | 2 | 7.0 | – | 7.3 | 19.1 | – | 23.0 | 7.1 | 21.0 | 0.2 | 2.8 | 7.3 | 23.0 |

**Table 6. Measurements [in µm] and *pt* values of selected morphological structures of individuals of *Ramazzottius syraxi* sp. nov. from the additional locality, mounted in Hoyer's medium (N – number of specimens/structures measured; RANGE refers to the smallest and the largest structure among all measured specimens; SD – standard deviation, *pt* – ratio of the length of a given structure to the length of the buccal tube expressed as a percentage,? – lack of measurements due to unsuitable position of the structure).**

| CHARACTER | N | RANGE | | | | | | MEAN | | SD | |
|---|---|---|---|---|---|---|---|---|---|---|---|
| | | µm | | | *pt* | | | µm | *pt* | µm | *pt* |
| Body length | 4 | 137 | – | 274 | | – | | 226 | | 64 | |
| Buccal tube | | | | | | | | | | | |
| Buccal tube length | 4 | 22.1 | – | 29.4 | | – | | 26.4 | – | 3.1 | – |
| Stylet support insertion point | 4 | 13.2 | – | 17.2 | 57.3 | – | 59.8 | 15.5 | 58.7 | 1.7 | 1.1 |
| Buccal tube external width | 4 | 1.8 | – | 2.2 | 7.3 | – | 8.3 | 2.0 | 7.7 | 0.2 | 0.4 |
| Buccal tube internal width | 4 | 0.8 | – | 0.9 | 2.9 | – | 3.8 | 0.8 | 3.2 | 0.0 | 0.4 |
| Placoid lengths | | | | | | | | | | | |
| Macroplacoid 1 | 4 | 2.1 | – | 3.0 | 9.3 | – | 10.7 | 2.7 | 10.0 | 0.4 | 0.6 |
| Macroplacoid 2 | 4 | 1.8 | – | 2.6 | 8.1 | – | 9.0 | 2.3 | 8.6 | 0.4 | 0.4 |
| Macroplacoid row | 4 | 4.0 | – | 6.5 | 18.2 | – | 22.4 | 5.5 | 20.7 | 1.1 | 1.9 |
| Claw i heights | | | | | | | | | | | |
| External base | 2 | 4.9 | – | 7.2 | 22.1 | – | 24.4 | 6.0 | 23.3 | 1.6 | 1.7 |
| External primary branch | 0 | | ? | | | ? | | ? | ? | ? | ? |
| External secondary branch | 2 | 4.6 | – | 7.4 | 20.7 | – | 25.1 | 6.0 | 22.9 | 2.0 | 3.1 |
| Internal base | 4 | 3.6 | – | 6.4 | 16.2 | – | 21.8 | 4.8 | 18.1 | 1.2 | 2.6 |
| Internal primary branch | 4 | 4.6 | – | 8.0 | 20.4 | – | 27.3 | 6.1 | 22.7 | 1.4 | 3.1 |
| Internal secondary branch | 4 | 4.0 | – | 5.4 | 17.6 | – | 20.0 | 4.9 | 18.6 | 0.6 | 1.1 |
| Claw ii heights | | | | | | | | | | | |
| External base | 3 | 4.7 | – | 9.0 | 21.2 | – | 30.5 | 7.1 | 26.4 | 2.2 | 4.8 |
| External primary branch | 3 | 6.6 | – | 10.5 | 30.0 | – | 36.1 | 9.0 | 33.9 | 2.1 | 3.4 |
| External secondary branch | 3 | 4.6 | – | 7.0 | 20.9 | – | 23.7 | 5.9 | 22.3 | 1.2 | 1.4 |
| Internal base | 4 | 3.7 | – | 6.3 | 15.4 | – | 21.3 | 4.8 | 18.0 | 1.1 | 2.6 |
| Internal primary branch | 4 | 5.0 | – | 8.1 | 21.4 | – | 27.4 | 6.3 | 23.8 | 1.3 | 2.6 |
| Internal secondary branch | 4 | 4.1 | – | 6.6 | 17.5 | – | 22.5 | 5.2 | 19.7 | 1.1 | 2.2 |
| Claw iii heights | | | | | | | | | | | |
| External base | 3 | 5.2 | – | 8.6 | 23.3 | – | 29.3 | 6.8 | 25.7 | 1.7 | 3.2 |
| External primary branch | 3 | 6.7 | – | 11.1 | 30.6 | – | 37.6 | 9.0 | 34.1 | 2.2 | 3.5 |
| External secondary branch | 3 | 4.2 | – | 6.6 | 18.9 | – | 22.5 | 5.5 | 20.7 | 1.2 | 1.8 |
| Internal base | 4 | 3.9 | – | 6.1 | 16.8 | – | 22.4 | 5.1 | 19.1 | 1.0 | 2.5 |
| Internal primary branch | 3 | 4.6 | – | 7.6 | 20.8 | – | 25.9 | 6.2 | 23.6 | 1.5 | 2.6 |
| Internal secondary branch | 4 | 3.6 | – | 7.0 | 16.3 | – | 23.9 | 5.1 | 19.1 | 1.4 | 3.5 |
| Claw iv lengths | | | | | | | | | | | |
| Anterior base | 1 | 4.5 | – | 4.5 | 20.2 | – | 20.2 | 4.5 | 20.2 | ? | ? |
| Anterior primary branch | 1 | 5.5 | – | 5.5 | 24.9 | – | 24.9 | 5.5 | 24.9 | ? | ? |
| Anterior secondary branch | 1 | 3.8 | – | 3.8 | 17.4 | – | 17.4 | 3.8 | 17.4 | ? | ? |
| Posterior base | 1 | 5.4 | – | 5.4 | 24.4 | – | 24.4 | 5.4 | 24.4 | ? | ? |
| Posterior primary branch | 0 | | ? | | | ? | | ? | ? | ? | ? |
| Posterior secondary branch | 1 | 4.2 | – | 4.2 | 19.1 | – | 19.1 | 4.2 | 19.1 | ? | ? |

Roszkowska, Gawlak, Draga & Kaczmarek 2019 [42]. *Macrobiotus hibiscus* was described from the type locality in Brazil [65] and later reported from Ecuador [84], Tanzania [85], New Zeland [86], Argentina [74,87,88], China [89,90], and USA (see review in Kaczmarek et al. [33]; [91]). But due to the extremely limited original description, the species was classified as *nomina inquirenda* by Stec et al. [18]. That is why all current records of *Mac. hibiscus* should be considered as doubtful, and the species require detailed examination of the type material or new specimens from the type locality. In turn, *Mac. polypiformis* and *Mac. dulciporus* were described from Ecuador recently and are known only from their type localities [40,42].

Regarding genus *Mesobiotus*, only two species have been recorded in Ecuador, i.e., *Meb. coronatus* and *Meb. romani* Roszkowska, Stec, Gawlak & Kaczmarek 2018 [41]. *Mesobiotus coronatus* was described from the type locality in Brazil [66] and later redescribed based on material from Ecuador [92]. This species has a relatively wide distribution: Costa Rica, Dominican Republic (Greater Antilles), Mexico (see review in Kaczmarek et al. [31]), Argentina, Chile, Peru, Uruguay (see review in Kaczmarek et al. [32]), USA (see review in Kaczmarek et al. [33]), New Zealand [77,86], Antarctica [93], Russia [94], China [95], Seychelles [96], Saint Martin (Lesser Antilles) [97]. Despite this, it was suggested by Pilato et al. [92] that true *Meb. coronatus* has confirmed localities only in South America. All other localities require a detailed reconsideration also since this species belongs to the *Mac. harmsworthi* group, which needs a really careful examination of specimens and eggs [55,69].

Specimens described by us as *Meb.* cf. *coronatus* differ from *Meb. coronatus* specimens, used in redescription of this species by Pilato et al. [92], by the size of the eggs, which can possibly suggest that our specimens belong to the new species. However, as showed by Kayastha et al. [98], size of the eggs in the same species can vary in the different localities. Taking into consideration this small morphological difference and still unclear taxonomic situation of *Meb. coronatus sensu lato*, i.e., i) lack of genetic data, ii) insufficient morphological description of the specimens from the type locality [66] and iii) morphological redescription based on specimens outside type locality [92], we decided not to propose a new taxon before integrative redescription of *Meb. coronatus sensu lato* based on specimens from the type locality in Brazil will be available. Additionally, it is probable that our population and specimens are redescribed as *Meb. coronatus* by Pilato et al. [92] belong to the same possible new taxon and will differ from *Meb. coronatus sensu lato* from type locality in Brazil.

*Mesobiotus romani* was described from Ecuador recently and is known only from the type locality [41].

*Ramazzottius syraxi* sp. nov. is the first species of the genus *Ramazzottius* found in Ecuador.

Therefore, the number of known species from mainland Ecuador has increased to 29 with two new described species and much more species is waiting for discovery in this region.

## Supporting information

**S1 Checklist. Questionnaire on inclusivity in global research.**
(PDF)

**S1 Table. Raw morphometric data for *Macrobiotus sharopovi* sp. nov.**
(XLSX)

**S2 Table. Raw morphometric data for *Mesobiotus* cf. *coronatus*.**
(XLSX)

**S3 Table. Raw morphometric data for *Ramazzottius syraxi* sp. nov. from the type locality.**
(XLSX)

**S4 Table. Raw morphometric data for *Ramazzottius syraxi* sp. nov. from the additional locality.**
(XLSX)

**S5 Table. Genetic distances for *Macrobiotus sharopovi* sp. nov.**
(XLSX)

**S6 Table. Genetic distances for *Mesobiotus* cf. *coronatus.***
(XLSX)

## Acknowledgments

We want to thank Pedro Rios Guayasamín from the Laboratorio de Ecología Tropical Natural y Aplicada, Universidad Estatal Amazónica (Puyo, Ecuador) and Milena Roszkowska for their help in the collection of samples. Also, we are grateful to Daniel Stec form Institute of Systematics and Evolution of Animals, Polish Academy of Sciences (Kraków, Poland) for kindly providing us help with analyzing of the DNA sequences.

Studies have been partially conducted in the framework of activities of BARg (Biodiversity and Astrobiology Research group).

## Author contributions

**Conceptualization:** Anastasiia Polishchuk.

**Data curation:** Anastasiia Polishchuk.

**Formal analysis:** Anastasiia Polishchuk, Pushpalata Kayastha, Dominika Młodzianowska, Martyna Michalska, Jędrzej Warguła, Łukasz Kaczmarek.

**Funding acquisition:** Anastasiia Polishchuk.

**Investigation:** Anastasiia Polishchuk, Pushpalata Kayastha, Dominika Młodzianowska, Martyna Michalska, Jędrzej Warguła, Łukasz Kaczmarek.

**Methodology:** Pushpalata Kayastha, Magdalena Gawlak, Jędrzej Warguła, Łukasz Kaczmarek.

**Project administration:** Łukasz Kaczmarek.

**Resources:** Magdalena Gawlak, Łukasz Kaczmarek.

**Supervision:** Łukasz Kaczmarek.

**Validation:** Łukasz Kaczmarek.

**Visualization:** Anastasiia Polishchuk, Pushpalata Kayastha, Dominika Młodzianowska, Martyna Michalska, Magdalena Gawlak.

**Writing – original draft:** Anastasiia Polishchuk, Pushpalata Kayastha, Dominika Młodzianowska, Martyna Michalska, Łukasz Kaczmarek.

**Writing – review & editing:** Anastasiia Polishchuk, Pushpalata Kayastha, Jędrzej Warguła, Łukasz Kaczmarek.

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
