## [Decision Letter · Decision Letter 0]

25 Feb 2025

PONE-D-25-06719

An integrative description of three new eutardigrade species from Cotacachi-Cayapas National Park, Ecuador

PLOS ONE

Dear Dr. Polishchuk,

Thank you for submitting your manuscript to PLOS ONE. After careful consideration, we feel that it has merit but does not fully meet PLOS ONE’s publication criteria as it currently stands. Therefore, we invite you to submit a revised version of the manuscript that addresses the points raised during the review process.

While the manuscript presents a significant effort to integrate morphological and molecular data, there are critical issues regarding two of the proposed species descriptions that need to be resolved before publication. See below:

**a) *Mesobiotus * species description:**the justification for describing this species is insufficient. The closest known species, *Mesobiotus coronatus* , has been reported in South America, but the lack of genetic data prevents a reliable comparison. The minor differences in egg morphometric traits are not strong enough to support species differentiation, especially given recent findings that question their reliability. The authors should either provide stronger evidence.

**b) *Ramazzottius * species description: **this species is described based solely on morphological data, as DNA sequencing was unsuccessful. While this is not uncommon in tardigrade research, the evidence presented does not convincingly support the designation of a new species. The gibbosities mentioned are not clearly visible in the provided images, necessitating improved photographs. Additionally, the closest morphological match appears to be *Ramazzottius baumanni* , not *R. saltensis* or *R. szeptycki* , as proposed by the authors. A revised differential diagnosis is needed, and if the species cannot be reliably distinguished from *R. baumanni* , its description should be reconsidered.

**c) Text length:** The text, mainly Introduction, contains redundant information and should be condensed. The section from lines 83 to 127, for example, can be reduced to a brief reference to previous reviews and key studies on the relevant genera.

I encourage the authors to undertake these revisions to enhance the manuscript’s scientific rigor and clarity before resubmission.

We look forward to receiving your revised manuscript.

Kind regards,

Wesley Dondoni Colombo

Academic Editor

PLOS ONE

Journal Requirements:

 “The work of Anastasiia Polishchuk was supported by grant no. 2022/01/4/NZ4/00009 from the National Science Centre (Poland)”

4. Please take this opportunity to be sure you have met all of our guidelines for new species. For proper registration of a new zoological taxon, we require two specific statements to be included in your manuscript.

1.        In the Results section, the globally unique identifier (GUID), currently in the form of a Life Science Identifier (LSID), should be listed under the new species name, for example:

Anochetus boltoni Fisher sp. nov. urn:lsid:zoobank.org:act:B6C072CF-1CA6-40C7-8396-534E91EF7FBB

Another LSID for the manuscript itself should also appear within the Nomenclature statement. You will need to contact Zoobank (zoobank.org/About) to obtain a GUID (LSID). You should receive one LSID for your manuscript and a separate, unique LSID for the new species.

2.        Please also insert the following text into the Methods section, in a sub-section to be called "Nomenclatural Acts":

The electronic edition of this article conforms to the requirements of the amended International Code of Zoological Nomenclature, and hence the new names contained herein are available under that Code from the electronic edition of this article. This published work and the nomenclatural acts it contains have been registered in ZooBank, the online registration system for the ICZN. The ZooBank LSIDs (Life Science Identifiers) can be resolved and the associated information viewed through any standard web browser by appending the LSID to the prefix "http://zoobank.org/". The LSID for this publication is: urn:lsid:zoobank.org:pub: XXXXXXX. The electronic edition of this work was published in a journal with an ISSN, and has been archived and is available from the following digital repositories: PubMed Central, LOCKSS [author to insert any additional repositories].

All PLOS ONE articles are deposited in PubMed Central and LOCKSS. If your institute, or those of your co-authors, has its own repository, we recommend that you also deposit the published online article there and include the name in your article.

Following a recent ruling by the International Commission on Zoological Nomenclature, electronic journals are now a valid format for publication of new zoological taxa. In order to ensure the valid publication of your new species, please be sure to include the updated version of Nomenclatural Acts (above). A complete explanation of our guidelines for publishing new species can be found on our website: http://www.plosone.org/static/guidelines#zoological.

Reviewers' comments:

Reviewer's Responses to Questions

**Comments to the Author**

1. Is the manuscript technically sound, and do the data support the conclusions?

Reviewer #1: Partly

Reviewer #2: Partly

2. Has the statistical analysis been performed appropriately and rigorously? 

Reviewer #1: N/A

Reviewer #2: N/A

3. Have the authors made all data underlying the findings in their manuscript fully available?

Reviewer #1: Yes

Reviewer #2: Yes

4. Is the manuscript presented in an intelligible fashion and written in standard English?

Reviewer #1: Yes

Reviewer #2: Yes

5. Review Comments to the Author

Reviewer #1: The manuscript “An integrative description of three new eutardigrade species from Cotacachi-Cayapas

National Park, Ecuador” attempts to describe three new eutardigrade species. However, for two of them I have serious concerns (see below). I didn’t check the English of the manuscript as I am not a native speaker, but I could easily understand everything so there are not issues from that point of view.

1) The introduction should be shortened, some parts are just repetitions of previous works and unnecessary: line 83 to 127 could be easily reduce to three lines referring to previous reviews or papers on the three genera mentioned.

2) The Macrobiotus description looks fine and I think it is a genuinely new species.

3) There are not the conditions to describe the Mesobiotus species: the most similar species (M. coronatus) has been reported (and described) from South America, but no DNA sequences are available for M. coronatus and thus a genetic comparison is not possible. The purported egg morphometric differences between the new species and M. coronatus are not strong enough to support the differentiation. In the paper “Kayastha, P., Szydło, W., Mioduchowska, M., & Kaczmarek, Ł. (2023). Morphological and genetic variability in cosmopolitan tardigrade species—Paramacrobiotus fairbanksi Schill, Förster, Dandekar & Wolf, 2010. Scientific Reports, 13(1), 17672.” the variability of egg morphometric traits highlight how those characters are not reliable for species differentiation.

4) In the description of Ramazzottius syraxi, the gibbosities mentioned by the authors are very poorly visible in the photographs, so at least the authors should provide more clear images. However the authors claim that the new species is similar to R. saltensis and R. szeptycki, while by comparing the authors photographs with the ones from “Dey, P. K., López-López, A., Morek, W., & Michalczyk, Ł. (2024). Tardigrade Augean stables—a challenging phylogeny and taxonomy of the family Ramazzottiidae (Eutardigrada: Hypsibioidea). Zoological Journal of the Linnean Society, 200(1), 95-110.”, it is clear that the m morphologically closest species is Ramazzottius baumanni (which is also a Neotropical species). The authors should carefully redo the differential diagnosis and reconsider if this species is really a new one or R. baumanni.

To summarize, I strongly suggest the authors to drop the Mesobiotus species description and carefully re-examine the Ramazzottius one (if it turns out to be indistinguishable from R. baumanni it should then be dropped too).

Reviewer #2: The submitted manuscript describes three new tardigrade species from Ecuador. The authors have attempted to collect comprehensive data within the framework of integrative taxonomy by combining morphological and molecular investigations. They were successful in two cases; however, the putative new Ramazzottius species was described based solely on morphological data due to difficulties in obtaining reliable DNA sequences. This is not uncommon when working with tardigrades, especially if the material is not very fresh. Overall, the data collected are of good quality, and their presentation is generally sufficient in most cases. However, I have identified some issues with data interpretation, leading me to recommend major revision.

The species of the genus Macrobiotus appear to be appropriately differentiated from similar taxa, and the erection of this species is justified. I have provided only minor comments on this species in the PDF file.

However, the erection of the new species within the genus Mesobiotus seems premature and not well justified at this time. The differential diagnosis includes two other highly similar taxa that cannot currently be easily or confidently distinguished from the newly studied population. I have elaborated on these issues more thoroughly in the PDF file. The morphology of the analyzed population fits well with the diagnosis of M. coronatus, with only small differences in egg metric characters—differences that many tardigradologists would not consider sufficient or reliable for species delineation. Unless stronger evidence is presented in favor of erecting a new species, I recommend that this data be published as a revision of M. coronatus. The data collected by the authors will still be very valuable to other researchers.

For the new species of Ramazzottius, it was challenging to assess its distinctiveness from related species due to the chaotic description and unclear presentation of the sculpture and gibbosity arrangements. I have provided suggestions that should help clarify the presentation and improve the interpretability of the data.

6. PLOS authors have the option to publish the peer review history of their article (what does this mean? ). If published, this will include your full peer review and any attached files.

**Do you want your identity to be public for this peer review?** For information about this choice, including consent withdrawal, please see our Privacy Policy .

Reviewer #1: No

Reviewer #2: No

---

## [Author Response · Author response to Decision Letter 1]

10 Apr 2025

Editor:

GENERAL COMMENTS:

a) Mesobiotus species description: the justification for describing this species is insufficient. The closest known species, Mesobiotus coronatus, has been reported in South America, but the lack of genetic data prevents a reliable comparison. The minor differences in egg morphometric traits are not strong enough to support species differentiation, especially given recent findings that question their reliability. The authors should either provide stronger evidence.

b) Ramazzottius species description: this species is described based solely on morphological data, as DNA sequencing was unsuccessful. While this is not uncommon in tardigrade research, the evidence presented does not convincingly support the designation of a new species. The gibbosities mentioned are not clearly visible in the provided images, necessitating improved photographs. Additionally, the closest morphological match appears to be Ramazzottius baumanni, not R. saltensis or R. szeptycki, as proposed by the authors. A revised differential diagnosis is needed, and if the species cannot be reliably distinguished from R. baumanni, its description should be reconsidered.

c) Text length: The text, mainly Introduction, contains redundant information and should be condensed. The section from lines 83 to 127, for example, can be reduced to a brief reference to previous reviews and key studies on the relevant genera.

REPLY: Detailed responses to all the above comments were provided to each of the reviews separately.

Review Comments

Reviewer 1:

GENERAL COMENT: The manuscript “An integrative description of three new eutardigrade species from Cotacachi-Cayapas National Park, Ecuador” attempts to describe three new eutardigrade species. However, for two of them I have serious concerns (see below). I didn’t check the English of the manuscript as I am not a native speaker, but I could easily understand everything so there are not issues from that point of view.

COMMENT 1: The introduction should be shortened, some parts are just repetitions of previous works and unnecessary: line 83 to 127 could be easily reduce to three lines referring to previous reviews or papers on the three genera mentioned.

REPLY: The text has been revised and shortened.

COMMENT 2: The Macrobiotus description looks fine and I think it is a genuinely new species.

REPLY: Thank you for your consideration.

COMMENT 3: There are not the conditions to describe the Mesobiotus species: the most similar species (M. coronatus) has been reported (and described) from South America, but no DNA sequences are available for M. coronatus and thus a genetic comparison is not possible. The purported egg morphometric differences between the new species and M. coronatus are not strong enough to support the differentiation. In the paper “Kayastha, P., Szydło, W., Mioduchowska, M., & Kaczmarek, Ł. (2023). Morphological and genetic variability in cosmopolitan tardigrade species—Paramacrobiotus fairbanksi Schill, Förster, Dandekar & Wolf, 2010. Scientific Reports, 13(1), 17672.” the variability of egg morphometric traits highlight how those characters are not reliable for species differentiation.

REPLY: We agree with the reviewer and modified the text of the manuscript. We decided to describe a population which we found as Meb. cf. coronatus with all details necessary for recognition of this species/population in the future. We agree that situation with Meb. coronatus is unclear. We still think that our species is new for science, but we will wait for integrative redescription of Meb. coronatus from the type locality. We added an explanation of the problematic taxonomic situation of Meb. coronatus to the Conclusions in our manuscript.

COMMENT 4: In the description of Ramazzottius syraxi, the gibbosities mentioned by the authors are very poorly visible in the photographs, so at least the authors should provide more clear images. However the authors claim that the new species is similar to R. saltensis and R. szeptycki, while by comparing the authors photographs with the ones from “Dey, P. K., López-López, A., Morek, W., & Michalczyk, Ł. (2024). Tardigrade Augean stables—a challenging phylogeny and taxonomy of the family Ramazzottiidae (Eutardigrada: Hypsibioidea). Zoological Journal of the Linnean Society, 200(1), 95-110.”, it is clear that the m morphologically closest species is Ramazzottius baumanni (which is also a Neotropical species). The authors should carefully redo the differential diagnosis and reconsider if this species is really a new one or R. baumanni.

REPLY: The description and figures were improved, the figures have been updated to indicate the structures (rows of transverse sculptured bands, gibbosities and concave areas). We also added more taxa to the morphological differential diagnosis (including populations described by Dey et al. 2024). However, we want to stress up here that these authors also did not redescribe Ram. baumanni integratively based on material from the type locality, and their populations may also be a separate species, as emphasized in their work. With the updates, additional figures, as well as the adding of Ram. baumanni species complex to the differential diagnosis and further comparisons, we demonstrated that our species is after all different from the suggested Ram. baumanni and other similar taxa.

COMMENT 5: To summarize, I strongly suggest the authors to drop the Mesobiotus species description and carefully re-examine the Ramazzottius one (if it turns out to be indistinguishable from R. baumanni it should then be dropped too).

REPLY: The species Meb. shevchenkoi sp. nov. was revised, and we decided to consider it as Mesobiotus cf. coronatus, as explained above. Description of new Ramazzottius was updated, as explained above, and we think that now it can be considered as new species even without DNA analyses.

Reviewer 2:

GENERAL COMENT: The submitted manuscript describes three new tardigrade species from Ecuador. The authors have attempted to collect comprehensive data within the framework of integrative taxonomy by combining morphological and molecular investigations. They were successful in two cases; however, the putative new Ramazzottius species was described based solely on morphological data due to difficulties in obtaining reliable DNA sequences. This is not uncommon when working with tardigrades, especially if the material is not very fresh. Overall, the data collected are of good quality, and their presentation is generally sufficient in most cases. However, I have identified some issues with data interpretation, leading me to recommend major revision.

COMMENT 1: The species of the genus Macrobiotus appear to be appropriately differentiated from similar taxa, and the erection of this species is justified. I have provided only minor comments on this species in the PDF file.

REPLY: All comments from the PDF file have been reviewed and we responded to all of them in the section below. All minor corrections have been added.

COMMENT 2: However, the erection of the new species within the genus Mesobiotus seems premature and not well justified at this time. The differential diagnosis includes two other highly similar taxa that cannot currently be easily or confidently distinguished from the newly studied population. I have elaborated on these issues more thoroughly in the PDF file. The morphology of the analyzed population fits well with the diagnosis of M. coronatus, with only small differences in egg metric characters—differences that many tardigradologists would not consider sufficient or reliable for species delineation. Unless stronger evidence is presented in favor of erecting a new species, I recommend that this data be published as a revision of M. coronatus. The data collected by the authors will still be very valuable to other researchers.

REPLY: We agree with the reviewer and modified the text of the manuscript. We decided to describe a population which we found as Meb. cf. coronatus with all details necessary for recognition of this species/population in the future. We agree that situation with Meb. coronatus is unclear. We still think that our species is new for science, but we will wait for integrative redescription of Meb. coronatus from the type locality. We also cannot redescribe Meb. coronatus based on our material, as the found specimens are far from the the type locality of this species. Such solution could possibly create another taxonomic problem with the status of Meb. coronatus. However, we added an explanation of the problematic taxonomic situation of Meb. coronatus to the Conclusions in our manuscript.

COMMENT 3: For the new species of Ramazzottius, it was challenging to assess its distinctiveness from related species due to the chaotic description and unclear presentation of the sculpture and gibbosity arrangements. I have provided suggestions that should help clarify the presentation and improve the interpretability of the data.

REPLY: The description and figures were improved, the figures have been updated to indicate the structures (rows of transverse sculptured bands, gibbosities and concave areas). We also added, more taxa to the morphological differential diagnosis (including populations described by Dey et al. 2024). However, we want to stress up here that these authors also did not redescribe Ram. baumanni integratively based on material from the type locality, and their populations may also be a separate species, as emphasized in their work. With the updates, additional figures, as well as the adding of Ram. baumanni species complex to the differential diagnosis and further comparisons, we demonstrated that our species is after all different from the suggested Ram. baumanni and other similar taxa.

Comments from the PDF file:

LINE 75: to support this statement please input reference [15] at the end.

REPLY: Corrected as suggested.

LINES 257–258: How granulation is arranged on the distal portion of the legs might serve as a valuable distinguishing characteristic. Therefore, I strongly suggest providing a detailed description of its distribution in the legs of the new species. Is granulation present on both the external and internal surfaces of legs I–III? If so, are these external and internal patches discontinuous, or are they connected by a thinner granulation band extending across the frontal leg surface? If the latter is the case, granulation in legs I–III might be continuous, with the external patch extending through the frontal surface to connect with the internal patch. What is the situation in the hind legs? Is granulation present on the latero-dorsal and dorsal surfaces of these legs? Please describe this with proper referencing to the figures if possible.

REPLY: Detailed description of the granulation was provided.

LINES 280–281: This statement cannot be supported by the provided photographic documentation. To me, it appears that the ventral portion also forms a continuous ridge with two thickenings. These thickenings might correspond to either two separate teeth or merely two tooth-like elevations within the ventral ridge. However, this characteristic cannot be stated with confidence without SEM documentation of the ventral portion of the OCA. Please revise this statement accordingly.

REPLY: This part was revised and corrected according to reviewer’s recommendations.

LINES 297–298: Include here what is their morphological state according to the reference you provided in M&M section e.g. divided cuticular bars etc.

REPLY: Corrected as suggested.

LINE 325: I suggest to remove this section as I do not see it as relevant or needed in the presented submission.

REPLY: Thank you for your suggestion, but we want to keep this part in the manuscript as well to supplement and interpret the raw data.

LINE 370: This species exhibit also proximal and distal patch of granulation on the leg IV. Such double granulation is lacking in your new species.

REPLY: Information was added to the text.

LINE 372: Provide numerical values or ranges. From the provided photographic documentation, it appears that most of the filaments are broken and cannot be easily counted. Therefore, I am concerned about the reliability of this distinguishing characteristic. If you can determine the range of filament numbers on the terminal disc for the new species, this should also be included in the egg description section.

REPLY: Information was added as suggested.

LINES 421–422: Please refer to my comment above regarding the granulation. Kindly check your specimens carefully and provide more detailed descriptions of the exact arrangement of granulation on all legs.

REPLY: Revised and corrected.

LINE 438: The ventral tooth/ridge is not almost round. I would say it is bar-shaped or rod-shape tooth.

REPLY: Corrected as suggested.

LINE 475: I cannot see dots in the photos provided. I would say that a dark sculpture is visible on the egg surface between processes and then that in SEM this sculpture seems to be caused by evident wrinkles of diverse shapes

REPLY: Text was modified and corrected as suggested.

LINE 480: I am not sure it can be stated that they are distributed regularly. Moreover, based on the SEM documentation, I am also unsure if these hollows correspond well with the internal 'reticulation' within the processes. To me, these appear to be two distinct structures and should be described separately. I concluded this because there are portions of the processes that are evidently smooth and lack these hollows, yet, thanks to the thin gold sputtering, the reticulum is still visible within their walls. Therefore, I would suggest that the 'reticulum' and the hollows are two distinct characteristics observable in the eggs of this species.

REPLY: We cannot agree with the reviewer. These hollows are emanation of internal reticulation. This is because the gold-layer on the mounted for SEM eggs was very thin, and some parts of the surface of the egg processes simply collapsed. Which means there is no sense to describe it as a separate character, especially since it is visible on some eggs and not on others. Also, in our opinion, the reticulation is quite regular without any specific patterns.

LINE 491: I suggest to remove this part as I don't see a strong relevance to the main outcome of this study. Thorp normalized data can be included in the supplementary materials with measurements.

REPLY: Thank you for your suggestion, but as stated above, we want to keep this part in the manuscript.

LINE 527: The surface is not reticulated, this is an internal structure, please rewrite.

REPLY: Corrected as suggested.

LINE 528: Please also consider: Mesobiotus silesiacus, Mesobiotus imperialis, Mesobiotus mandalori. Moreover there were three species described recently from Sweden with quite common genral morphology and they may also fall within the criteria used for differential diagnosis.

REPLY: Since we accept the point of view that this species should be classified as Mesobiotus cf. coronatus, we have removed the part with the morphological differential diagnosis as not necessary in this case.

LINE 533: The two metric characters for the eggs seem to be a very weak differentiating feature. In light of recent reports highlighting significant variability in egg morphometrics and morphology within Macrobiotoidea (especially for Mesobiotus and Paramacrobiotus), the discrimination presented here cannot be easily trusted. Moreover, this species has been reported several times from Ecuador, making it likely that the population examined in this study also represents M. coronatus. Given the data, describing a new species based solely on these small differences is risky and unsupported. I suggest that the authors present the data for this population as a revised description of M. coronatus rather than erecting a new species. This should be accompanied by a brief discussion or remarks section addressing any potential taxonomic uncertainties associated with this particular taxon.

REPLY: We agree with the reviewer and modified the text of the manuscript. We decided to describe a population which we found as Meb. cf. coronatus with all details necessary for recognition of this species/population in the future. In this situation, morphological differen

---

## [Decision Letter · Decision Letter 1]

27 Apr 2025

Description of two new species and a new population of Mesobiotus cf. coronatus from Cotacachi-Cayapas National Park, Ecuador

PONE-D-25-06719R1

Dear Dr. Polishchuk,

We’re pleased to inform you that your manuscript has been judged scientifically suitable for publication and will be formally accepted for publication once it meets all outstanding technical requirements.

Kind regards,

Wesley Dondoni Colombo

Academic Editor

PLOS ONE

Additional Editor Comments (optional):

Reviewers' comments:

Reviewer's Responses to Questions

**Comments to the Author**

1. If the authors have adequately addressed your comments raised in a previous round of review and you feel that this manuscript is now acceptable for publication, you may indicate that here to bypass the “Comments to the Author” section, enter your conflict of interest statement in the “Confidential to Editor” section, and submit your "Accept" recommendation.

Reviewer #2: All comments have been addressed

2. Is the manuscript technically sound, and do the data support the conclusions?

Reviewer #2: Yes

3. Has the statistical analysis been performed appropriately and rigorously? 

Reviewer #2: N/A

4. Have the authors made all data underlying the findings in their manuscript fully available?

Reviewer #2: Yes

5. Is the manuscript presented in an intelligible fashion and written in standard English?

Reviewer #2: Yes

6. Review Comments to the Author

Reviewer #2: I would like to express my thanks to the authors for such a detailed revision. In my opinion, the current version of the manuscript is suitable for publication. The two new tardigrade taxa are clearly differentiated from similar species, supporting their status as distinct species. The cautious approach taken with the Mesobiotus population, reported as M. cf. coronatus, is much appreciated. Although this population cannot currently be described as a new species, the data collected by the authors will be very valuable to the community.

I carefully reviewed the revised submission and found no further issues, except for one small detail regarding the supplementary materials. The Excel file with morphometric data for Mesobiotus populations still uses the provisional name with sp. nov. and should be updated to M. cf. coronatus to avoid confusion. This is a minor and easily correctable point.

With that adjustment, I consider this submission ready for publication. Many thanks to the authors for their hard work on tardigrade taxonomy in the tropics!

7. PLOS authors have the option to publish the peer review history of their article (what does this mean? ). If published, this will include your full peer review and any attached files.

**Do you want your identity to be public for this peer review?** For information about this choice, including consent withdrawal, please see our Privacy Policy .

Reviewer #2: No

---

## [Editor Report · Acceptance letter]

PONE-D-25-06719R1

PLOS ONE

Dear Dr. Polishchuk,

I'm pleased to inform you that your manuscript has been deemed suitable for publication in PLOS ONE. Congratulations! Your manuscript is now being handed over to our production team.

Kind regards,

on behalf of

Dr. Wesley Dondoni Colombo

Academic Editor

PLOS ONE